# Small RNA-Seq Transcriptome Profiling of Mesothelial and Mesothelioma Cell Lines Revealed microRNA Dysregulation after Exposure to Asbestos-like Fibers

**DOI:** 10.3390/biomedicines11020538

**Published:** 2023-02-13

**Authors:** Veronica Filetti, Claudia Lombardo, Carla Loreto, George Dounias, Massimo Bracci, Serena Matera, Lucia Rapisarda, Venerando Rapisarda, Caterina Ledda, Ermanno Vitale

**Affiliations:** 1Occupational Medicine, Department of Clinical and Experimental Medicine, University of Catania, 95123 Catania, Italy; 2Human Anatomy and Histology, Department of Biomedical and Biotechnological Sciences, University of Catania, 95123 Catania, Italy; 3Human Anatomy, Department of Medical and Surgical Sciences and Advanced Technologies, University of Catania, 95123 Catania, Italy; 4Department of Occupational and Environmental Health, University of West Attica, 10563 Athens, Greece; 5Occupational Medicine, Department of Clinical and Molecular Sciences, Polytechnic University of Marche, 60126 Ancona, Italy

**Keywords:** malignant mesothelioma, asbestos-like fibers, fluoro-edenite, small RNA-Seq transcriptome profiling, biomarkers

## Abstract

Environmental exposure to fibers of respirable size has been identified as a risk for public health. Experimental evidence has revealed that a variety of fibers, including fluoro-edenite, can develop chronic respiratory diseases and elicit carcinogenic effects in humans. Fluoro-edenite (FE) is a silicate mineral first found in Biancavilla (Sicily, Italy) in 1997. Environmental exposure to its fibers has been correlated with a cluster of malignant pleural mesotheliomas. This neoplasm represents a public health problem due to its long latency and to its aggression not alerted by specific symptoms. Having several biomarkers providing us with data on the health state of those exposed to FE fibers or allowing an early diagnosis on malignant pleural mesothelioma, still asymptomatic patients, would be a remarkable goal. To these purposes, we reported the miRNA transcriptome in human normal mesothelial cell line (MeT-5A) and in the human malignant mesothelioma cell line (JU77) exposed and not exposed to FE fibers. The results showed a difference in the number of deregulated miRNAs between tumor and nontumor samples both exposed and not exposed to FE fibers. As a matter of fact, the effect of exposure to FE fibers is more evident in the expression of miRNA in the tumor samples than in the nontumor samples. In the present paper, several pathways involved in the pathogenesis of malignant pleural mesothelioma have been analyzed. We especially noticed the involvement of pathways that have important functions in inflammatory processes, angiogenesis, apoptosis, and necrosis. Besides this amount of data, further studies will be designed for the selection of the most significant miRNAs to test and validate their diagnostic potential, alone or in combination with other protein biomarkers, in high-risk individuals’ liquid biopsy to have a noninvasive tool of diagnosis for this neoplasm.

## 1. Introduction

Malignant pleural mesothelioma (MPM) is an aggressive neoplasm of the pleural surface causally related to exposure to asbestos fibers [1]. In fact, several studies conducted in Finland [2], California, USA [3], China [4], Corsica [5], New Caledonia [6], Cyprus [7], and Greece [8] testified to a high incidence of MPM due to asbestos exposure. Yet, literature has now made it known that even exposure to asbestos-like fibers causes MPM [9]. Fluoro-edenite (FE), along with magnesio-riebeckite, erionite, winchite, antigorite, richterite, and Libby asbestos fibers, fall among these [9]. FE is an amphibole, found in Biancavilla (Sicily, Italy), a small town of the Etnean volcanic complex (Sicily, Italy) [10,11]. This mineral has some characteristics similar to the asbestos group [12] and has been classified as Group 1 human carcinogens [13].

Environmental exposure to these carcinogen fibers represents a public health problem due to the aggression of MPM not alerted by specific symptoms and to its long latency. It is clear that the first step to prevent carcinogen fibers exposure-related diseases is to reduce the presence of such airborne fibers through confinement, encapsulation, and reclamation of the artifacts that contain them [9]. These fibers are now ubiquitously present in the environment; therefore, having available biomarkers that give information on the health state of fibers-exposed subjects or that allow an early diagnosis on still asymptomatic MPM patients would be a great goal. Some studies have been conducted to evaluate the link between genetic variations in the molecular pathways and cancer risk to find useful biomarkers for screening and early diagnosis of MPM in fiber-exposed subjects [14,15,16,17]. Interestingly, the microRNAs (miRNAs) have been chosen by the scientific literature to be used both as valuable noninvasive diagnostic and prognostic biomarkers and as therapeutic targets for cancer [18,19].

As is widely known, miRNAs are evolutionarily conserved long 18–25 nucleotide (nt) single-stranded RNAs that negatively modulate the expression of their target mRNAs. miRNAs are important molecules in gene expression regulation at the post-transcriptional level, indeed, a single miRNA can control several mRNAs’ expression, and a single mRNA may be targeted by more than one miRNA, thus creating a network effect of cooperative regulation [20]. Many studies have been published about miRNAs and their involvement in the pathogenesis of several human cancers [21]. In addition, many efforts have been made to use circulating miRNAs as noninvasive diagnostic and prognostic biomarkers for several cancer types [22]. Unfortunately, despite the extensive work, to date, very few miRNAs are used in clinical practice [22].

As mentioned above, in this study, we reported a small RNA-Seq transcriptome profiling of human normal mesothelial cell line (MeT-5A) and in human malignant mesothelioma cell line (JU77) exposed and not exposed to FE fibers. Both these cell lines have been processed with and without FE fibers exposure, as reported in the experimental workflow (Figure 1). Subsequently, several pathways involved in the pathogenesis of MPM have been analyzed.

## 2. Materials and Methods

### 2.1. Cell Cultures

A normal human mesothelial cell line (MeT-5A) and a malignant human mesothelioma cell line (JU77) were obtained from the American Type Culture Collection (ATCC; Manassas, VA, USA). Both cell lines have been cultured in Roswell Park Memorial Institute 1640 (RPMI 1640) medium supplemented with 10% fetal bovine serum, 1% L-glutamine (Lonza; Walkersville, MD, USA), 1% non-essential amino acids solution (Gibco by Thermo Fisher, Waltham, MA, USA), and 1% penicillin/streptomycin (Lonza; Walkersville, MD, USA). The culture conditions were 37 °C in a humidified atmosphere with 5% CO_2_. The MeT-5A and JU77 cell lines were split 1:3 and 1:6, respectively, twice a week.

Cells in confluent condition were separated from the culture flask (SPL Life Sciences; Korea) using 0.25% trypsin in 2.21 mM EDTA solution (Corning; Manassas, VA, USA) and counted using Bürker chamber by Trypan Blue Stain 0.4% (Gibco by Life Technologies; New York, NY, USA). The cells used for the experiments were between II/III passages.

### 2.2. In Vitro Treatments

FE fibers were obtained from the Biancavilla quarry (Sicily, Italy). These were sampled using magnifiers, needles and tweezers. Subsequently the fibers were weighed, sterilized under UV light for 10 min, suspended in a known volume of RPMI 1640 medium, and sonicated through Omni-Ruptor 4000 Ultrasonic Homogenizer (OMNI International Inc.; Kennesaw, GA, USA) for 10 min. The stock solution was then diluted appropriately to obtain the different concentrations for in vitro treatments.

The in vitro functional experiments were preceded by the determination of the dose-response curves for both cell lines. MeT-5A were plated onto 96-well plates (Thermo Fisher Scientific; Roskilde, Denmark) for the dose-response curve at the density of 6 × 10^3^ cells/50 μL, while JU77 were plated at the density of 4 × 10^3^ cells/50 μL. After 24 h of incubation, 50 μL of FE fibers solutions were added to the cell cultures in amounts corresponding to final concentrations from 200 to 0.78 μg/mL. Both cell lines grown in FE-free medium were used as controls. At each time point (from 6 to 72 h of FE exposure) in cell culture, 10% MTT in Dulbecco’s Phosphate-Buffered Saline (DPBS) (Corning; Manassas, VA, USA) has been added to each well. After 4 h of incubation, the lysis solution (0.1% HCl conc. in absolute isopropyl alcohol) has been added to each well. The optical density was measured with an absorbance microplate reader (TECAN Trading AG; Switzerland) at λ = 620 nm. For each sample, three replicates were performed. Cell viability was calculated as the percentage of viable cells treated with FE fibers vs. untreated control cells as follows: Cell viability (%) = [OD (Treatment) − OD (Blank)]/[OD (Control) − OD (Blank)] ×100 [10]. Results have been analyzed using PRISM GraphPad 7.00 and data were represented as the mean ± SD. An unpaired Student’s t-test was used to compare data between the two groups. A value of *p* < 0.05 was considered statistically significant.

MeT-5A was plated at the density of 1 × 10^6^ cells while JU77 was plated at the density of 8.5 × 10^5^ cells onto 100 × 20 mm Petri Dishes (Eppendorf; Hamburg, Germany) for these functional experiments. After 24 h-incubation, the medium of both cell lines has been replaced with FE fibers solutions to final concentrations of 50 and 10 μg/mL [23]. MeT-5A and JU77 cells grown in normal medium were used as controls. After 48 h from FE fibers exposure, pellets were collected in duplicate.

After removing the supernatant, cells were harvested on ice by scraping in cold DPBS. Cells are then centrifuged at 0.2× *g* for 5 min at 4 °C and suspended in 1 mL cold DPBS. The cell solution was transferred to Eppendorf tubes and cells were centrifuged at 0.8× *g* for 5 min at 4 °C [23]. After removing the supernatant, samples were stored to −80 °C until RNA isolation.

### 2.3. RNA Isolation

Total RNA containing small non-coding RNA was extracted from the cell lines using miRNeasy Mini Kit (QIAGEN; Venlo, The Netherlands) according to the manufacturer’s recommended protocols (miRNeasy Mini Handbook 11/2020). The RNA was quantified by the absorbance ratio at λ = 260/280 nm through NanoDrop (ND 1000) UV–Vis spectrophotometer [12]. All samples were diluted at the final concentration of 50 ng/µL for the Small RNA-Seq.

### 2.4. Small RNA-Seq

The QIAseq miRNA library kit (QIAGEN, Hilden, Germany) was used for small RNA-Seq library preparation following the manufacturer’s instructions. RNA samples were quantified and quality tested by Agilent 2100 Bioanalyzer RNA assay (Agilent Technologies, Santa Clara, CA, USA). Libraries were then checked with both Qubit 2.0 Fluorometer (Invitrogen, Carlsbad, CA, USA) and Agilent Bioanalyzer DNA assay or Caliper (PerkinElmer, Waltham, MA, USA). Finally, libraries were prepared for sequencing and sequenced on single-end 150 bp mode on NovaSeq 6000 (Illumina, San Diego, CA, USA) [23].

### 2.5. Small RNA-Seq Analysis

Low-quality reads and adapters were trimmed using Trim Galore, which is a wrapper of FASTQC [24] and Cutadapt [25]. After that, trimmed reads were aligned onto the reference human genome (HG38 version) using HISAT2 [26]. The generated SAM files were first converted into BAM files using Samtools; secondly, mapped reads were quantified by feature Counts (parameters: -d 14 --primary) using a custom GTF annotation file containing the genomic coordinates of miRNAs (reported in miRBase [27]). The obtained raw count values were normalized to scale the raw library sizes in trimmed mean of M values (TMM) by using edgeR [28] and all miRNAs, whose geometric mean of TMM values across all samples was less than one, were removed from the analysis, because they were non-expressed or expressed at a very low level. Finally, the filtered count matrix was used for the differential expression analysis using LIMMA [29]. miRNA with a |Log2FC| > 0.58 and an adjusted *p*-value (Benjamini–Hochberg correction) <0.05 were considered differentially expressed. Finally, the impact of differentially expressed miRNAs on biological pathways was evaluated by using MITHrIL [30]. In this case, we selected all miRNAs with significant adjusted *p*-values without a Log2FC cutoff in order to evaluate also the impact of slightly differentially expressed miRNAs on biological pathways. All the above-mentioned steps of the analysis were performed using RNAdetector [31] software.

## 3. Results

### 3.1. Choice of Exposure Time

The in vitro functional experiments were preceded by the determination of the dose-response curves for both cell lines. The results showed a significant difference between 6 h treatments and all other time points (24, 48, and 72 h) at all tested concentrations exposure to FE fibers. Finally, an exposure time to the fibers equal to 48 h was chosen because longer exposure times did not cause statistically significant differences in cell viability (Figure 2 and Figure 3).

### 3.2. Differential Expression Analysis

In order to identify dysregulation in miRNAs induced by FE fibers, we performed an RNA-Seq transcriptome profiling of unexposed and exposed normal mesothelial (MeT5A) and malignant mesothelioma (JU77) cell lines. The differentially expressed miRNAs were: 333 untreated JU77 vs. MeT5A (145 up-regulated and 188 down-regulated), 323 in JU77 vs. MeT5A exposed to 10 μg/mL FE fibers (101 up-regulated and 222 down-regulated), and 325 in JU77 vs. MeT5A exposed to 50 μg/mL FE fibers (124 up-regulated and 201 down-regulated) (Table 1). Among these, 14 were in common between the JU77 and MeT5A untreated and exposed to 10 μg/mL FE fibers (6 up-regulated and 8 down-regulated), 21 were in common between the JU77 and MeT5A untreated and exposed to 50 μg/mL FE fibers (18 up-regulated and 3 down-regulated), and 19 were in common between the JU77 and MeT5A exposed to 10 and 50 μg/mL FE fibers (2 up-regulated and 17 down-regulated). Among all samples, the differentially expressed miRNAs in common were 54 (Table 2).

However, when we compared miRNAs expression between unexposed vs. exposed MeT-5A, the results showed very few differentially expressed molecules. In particular, there were 40 differentially expressed miRNAs in MeT5A untreated vs. exposed to 10 μg/mL FE fibers (22 up-regulated and 18 down-regulated) and 35 differentially expressed miRNAs in MeT5A untreated vs. exposed to 50 μg/mL FE fibers (8 up-regulated and 27 down-regulated). On the other hand, the comparison between untreated vs. FE-treated JU77 showed several differentially expressed miRNAs. In particular, miRNAs differentially expressed in JU77 unexposed vs. exposed to 10 and 50 μg/mL FE fibers were 113 (24 up-regulated and 89 down-regulated) and 125 (22 up-regulated and 103 down-regulated), respectively (Table 3). The miRNAs differentially expressed in common among these analyzed samples were not many. Samples showing more differentially expressed miRNAs in common were JU77 untreated and exposed to 10 μg/mL and 50 μg/mL FE fibers. In particular, these were 27 (1 up-regulated and 26 down-regulated). Non-neoplastic samples did not show differentially expressed miRNAs in common (Table 4). Among all samples, the differentially expressed miRNAs in common were 29, because hsa-miR-1248 was down-regulated comparing MeT10 vs. MeT5ANT and MeT50 vs. MeT5ANT and JU10 vs. JUNT samples and hsa-miR-3618 was down-regulated comparing MeT50 vs. MeT5ANT and JU10 vs. JUNT samples.

### 3.3. Pathways Analysis

Once the differentially expressed miRNAs for each comparison were identified, we investigated the impact of their dysregulation in metabolic and signaling pathways by using MITHrIL [30]. MITHrIL fully exploits the topological information encoded by pathways when computing perturbation scores. Pathways are then modeled as complex graphs where each node is a biological element (protein-coding gene, miRNA, or metabolite) and each edge is an interaction between them [30]. Importantly, MITHrIL takes into account experimentally validated miRNA-mRNA interactions so as to predict their effects on biological pathways [30].

The results demonstrated clear patterns of positive and negative perturbation scores involving 96 different pathways between the JU77 and MeT5A untreated and exposed to 10 and 50 μg/mL FE fibers. Of these, 28 showed mildly positive perturbation scores, while 68 showed mildly strong negative perturbation scores. Among these latter pathways, strong, negative patterns were observed in 25 pathways, including apoptosis; regulation of lipolysis in adipocytes; endocytosis; ovarian steroidogenesis; focal adhesion; progesterone-mediated oocyte maturation; axon guidance; adherence junction; longevity regulating pathway; regulation of actin cytoskeleton; sphingolipid signaling pathway; chemokine signaling pathway; leukocyte trans-endothelial migration; signaling pathways regulating pluripotency of stem cells; and specific signaling pathways of Rap 1, cGMP-PKG, AMPK, Hippo, PI3K-Akt, TGF-beta, ErbB, cAMP, Ras, and FoxO (Figure 4).

The pathway analysis performed between untreated vs. FE fibers treated JU77 and MeT-5A and the main impacted pathways showed clear patterns of positive and negative correlations involving 121 different pathways. Of these, 66 showed mildly strong, positive correlations and, among these 25, showed strong, positive correlations including signaling pathways of FoxO, PI3K-Akt, Rap1, VEGF, p53, and Ras; the regulation of actin cytoskeleton; signaling pathways of chemokine, prolactin, sphingolipid, estrogen, insulin, thyroid hormone, and oxytocin; long-term depression; progesterone-mediated oocyte maturation; natural killer cell-mediated cytotoxicity; Gap junction; serotonergic and cholinergic synapse; axon guidance; signaling pathways regulating the pluripotency of stem cells; longevity regulating pathways; and focal adhesion. Mildly strong, negative correlations were observed for 55 patterns, and among these, 14 showed strong, negative correlations, including metabolic pathways of arginine and proline, alanine, aspartate, and glutamate; glycolysis/gluconeogenesis; signaling pathways of adipocytokine; NF-kappa B; Toll-like receptor; glucagon; HIF-1; Jak-STAT; osteoclast differentiation; glycosaminoglycan degradation; cytokine–cytokine receptor interaction; and the adherence junction. The correlations between these pathways were quite obvious for the JU77 cell line untreated vs. treated with FE fibers (Figure 5).

## 4. Discussion

Small RNA-Seq transcriptome profiling of healthy mesothelium and MPM in vitro has been evaluated to highlight the deregulated miRNAs and the various pathways involved in an aggressive cancer such as MPM.

Certainly, the results showed a big difference in the number of deregulated miRNAs between tumor and nontumor samples both exposed and not exposed to FE fibers. As a matter of fact, the effect of exposure to FE fibers is more evident in the expression of miRNAs in tumor samples than in nontumor samples.

In the present paper, we analyzed several pathways that are involved in the pathogenesis of MPM. It is very interesting to point out the strong perturbation scores involving the above-mentioned pathways in the MPM vs. healthy mesothelial cell line. Among these, certain pathways were involved that play important roles in inflammatory processes and angiogenesis. Inflammation has a central role, since mesothelioma is a multicentric neoplasm that originates from inflammatory foci. Inflammation has been correlated with cancer, enhancing the development of malignancies [32]. In particular, the chemokine and TGF-beta signaling pathways lead to acute and chronic inflammation, the latter resulting in several fiber-associated pulmonary and pleural diseases [16,33]. The inflammasome is responsible for the activation of inflammatory processes via multiple mechanisms [33] that trigger a cell death process called proptosis, characterized both by apoptosis and necrosis. Cell death mechanisms and the release of chemokines and cytokines may help cancer regress and resist toxicity by fibers and cell growth [34] in inflammasome-dependent and -independent pathways [35]. Indeed, apoptosis after exposure to FE fibers is a mechanism meant to remove irreparably damaged cells, which causes genetic changes that predispose cells to neoplastic transformation [36]. Specific signaling pathways such as sphingolipid, FoxO, and Hippo pathways are involved in many important signal transduction processes, such as cell proliferation and apoptosis [37,38,39]. Dysregulation of the Hippo signaling pathway is highly conserved by phosphorylating and inhibiting the transcription coactivators YAP and TAZ, key regulators of proliferation and apoptosis. On the contrary, dephosphorylated YAP/TAZ moves into the nucleus and activates gene transcription through binding to the TEAD family and other transcription factors. Such changes in gene expression promote cell proliferation and stem cell/progenitor cell self-renewal but inhibit apoptosis, thereby promoting tissue regeneration, and tumorigenesis [38]. An experimental model demonstrated the activation of YAP caused by ATG7 deletion [40], which is an important transcription activator in malignant mesothelioma [41]. In our recent research, ATG7’s high expression represents a promising prognostic tool for patients with MPM [42]; thus, it would be interesting to explore whether there is an inverse correlation between ATG7 and YAP in malignant mesothelioma. The leukocyte migration to the site of injury is orchestrated by chemokines. Neutrophils are the first to be recruited onto the injury site, followed by monocytes, which differentiate into macrophages. Once activated, macrophages are the main source of growth factors and cytokines that affect the local microenvironment. Mast cells also contribute to inflammatory mediators, such as histamine, cytokines, and proteases, as well as lipid mediators [43]. In previous studies of our research group [16,44], we demonstrated the involvement of cytokines IL-18 and IL-1beta in the inflammasome activation process, suggesting that these immune modulators are involved in the pathogenic mechanisms triggered by FE fibers. Acute and chronic inflammations often generate common molecular mediators [37]. The strong inflammation, with an even increased angiogenesis, causes the fatal outcome of the neoplasm. The release of angiogenic cytokines, including TGF-beta and VEGF, occurs during the angiogenesis process in the malignant mesothelioma progression [45]. In particular, VEGF represents the main angiogenic cytokine involved in this cancer [46,47], modulating also the development of pleural effusion and ascites through an enhanced vascular permeability [47]. VEGF activation plays an essential role in increasing the survival of normal cells exposed to carcinogenic agents. FE fibers are able to induce functional modifications of parameters with crucial roles in cancer development and progression [11]. The synthesis of VEGF and beta-catenin, two critical steps of epithelial cell activation pathways, is affected by FE fibers exposure shown by an abnormal cellular status with up-regulated cell activities and a risk of neoplastic transformation [48]. Furthermore, the influence of FE fibers on cell motility has been demonstrated through a dysregulated and altered distribution of actin network [48]. Focal adhesion, which forms mechanical links between the cytoskeleton and extracellular matrix (ECM) and adherence junction, are clearly involved in malignant mesothelioma pathogenesis. About that, our recent study on FE exposure in lung fibroblasts suggested an ECM remodeling that can give rise to profibrotic cellular phenotypes and the tumor microenvironment [49]. The signaling pathway of natural killer (NK) cells mediated cytotoxicity has been shown to be involved in the development of malignant mesothelioma. However, the absence of NK cells does not alter tumor growth rates, suggesting that they cannot function as effector cells in this microenvironment. However, after local IL-2 and/or anti-CD40 antibody treatment of mesothelioma tumors, NK cells help acquire and/or maintain systemic immunity and long-term effector/memory responses [50].

Specific signaling pathways have been found to be involved in malignant mesothelioma. Among these, Ras and p53, the commonly mutated genes associated with cancer, are rarely targeted in malignant mesothelioma [51]. Ras has not been shown to be altered in mesothelioma cell lines [52,53]. However, several receptor tyrosine kinase pathways have been shown to be triggered in mesothelioma, including the epidermal growth factor receptor (EGFR), insulin-like growth factor receptor (IGFR), and c-Met [54,55,56], all of which activate Ras signaling. According to these results, several studies have already suggested that the PI3K-Akt pathway is hyperactivated in mesothelioma cell lines [51,57], resulting in the gain or loss of function of its downstream proteins, 4E-BP1 and pS6, both crucial to the regulation of protein synthesis [58]. However, the prognostic role of the PI3K pathway in MPM is not as yet quite defined [59].

## 5. Conclusions

Our goal was the validation of the most promising results in a subset of patients chronically exposed to FE fibers, using the liquid biopsy, to provide a minimally invasive screening tool for the secondary prevention of MPM. Early detection of circulating tumor biomarkers represents one of the most promising strategies to enhance the survival of cancer patients by increasing treatment efficiency [60]. Besides this large amount of data, further studies will be necessary in order to select the most significant miRNAs to test and validate their diagnostic potential, alone or in combination with other protein biomarkers, in high-risk individuals.

## Figures and Tables

**Figure 1 biomedicines-11-00538-f001:**
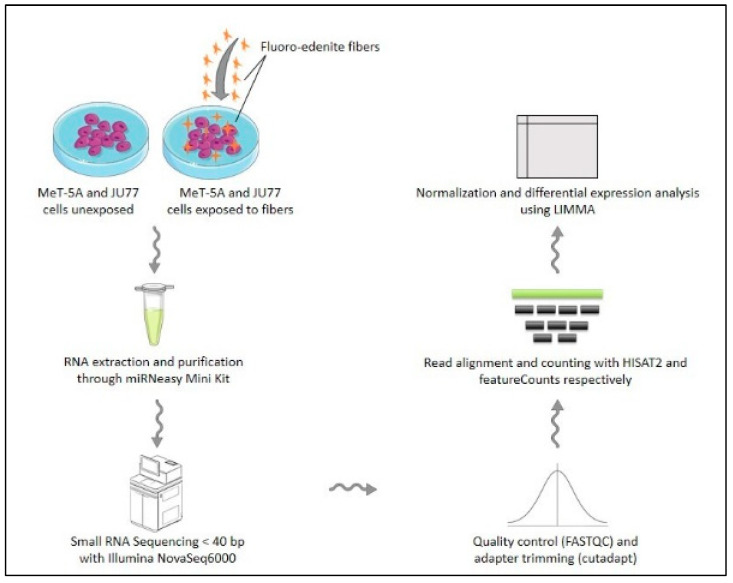
Experimental workflow.

**Figure 2 biomedicines-11-00538-f002:**
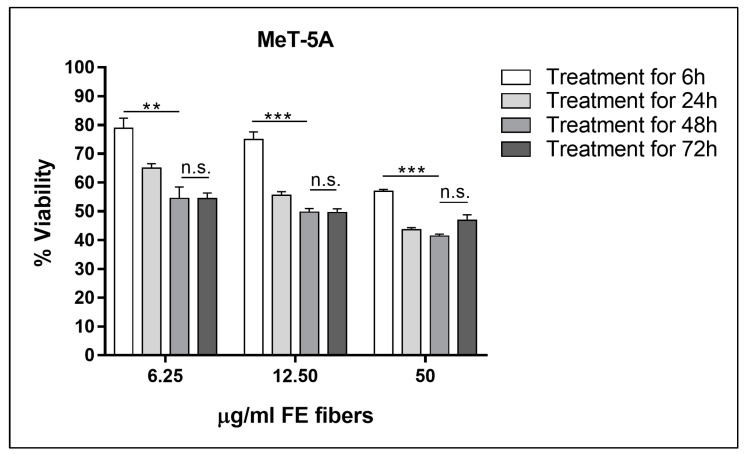
Cell viability of MeT-5A as a function of the exposure time course to the fluoro-edenite fibers at the concentrations of 6.25, 12.50, and 50 µg/mL (concentrations close to the IC_50_ values obtained after 48 h of treatments). Significance: n.s. not significant; ** *p* < 0.005; *** *p* < 0.0005.

**Figure 3 biomedicines-11-00538-f003:**
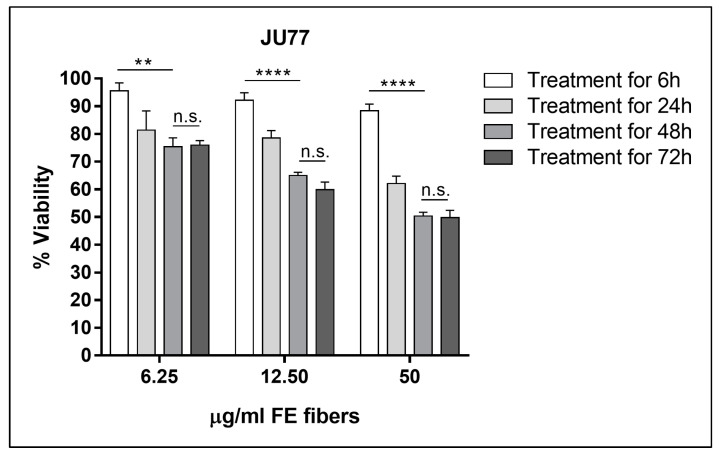
Cell viability of JU77 as a function of the exposure time course to the fluoro-edenite fibers at the concentrations of 6.25, 12.50, and 50 µg/mL (concentrations close to the IC_50_ values obtained after 48 h of treatments). Significance: n.s. not significant; ** *p* < 0.005; **** *p* < 0.00005.

**Figure 4 biomedicines-11-00538-f004:**
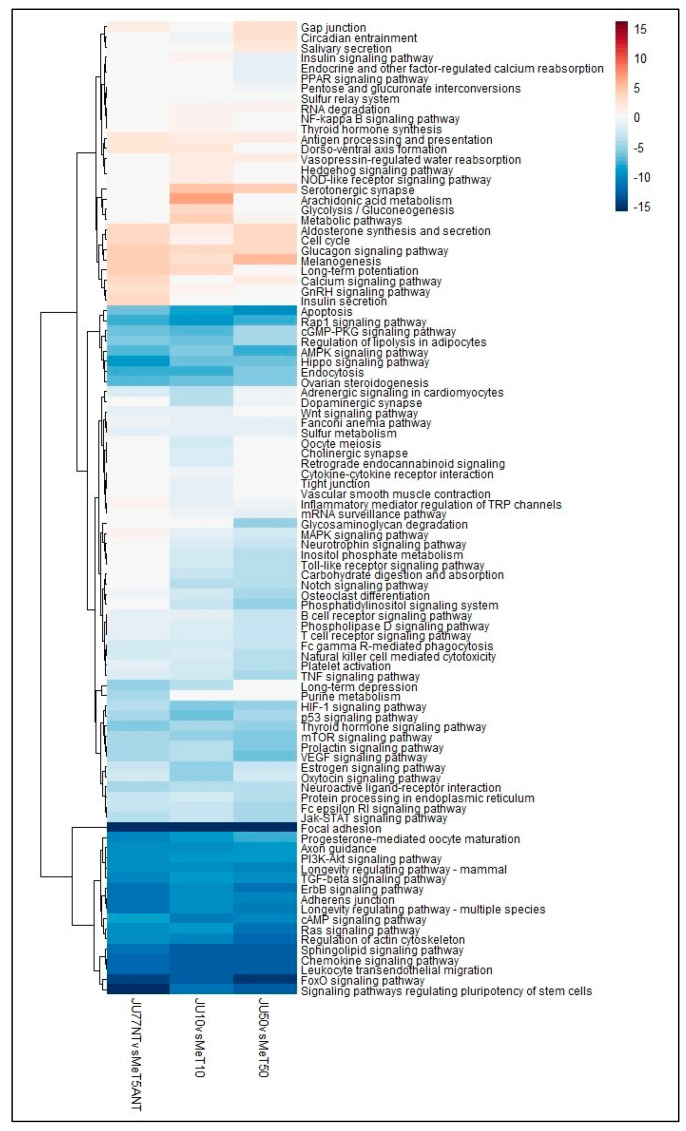
Heatmap showing the dysregulated pathways in JU77 (human malignant mesothelioma) vs. MeT5A (human normal mesothelium) at the same concentration of FE fibers exposure by taking into account the differentially expressed miRNAs.

**Figure 5 biomedicines-11-00538-f005:**
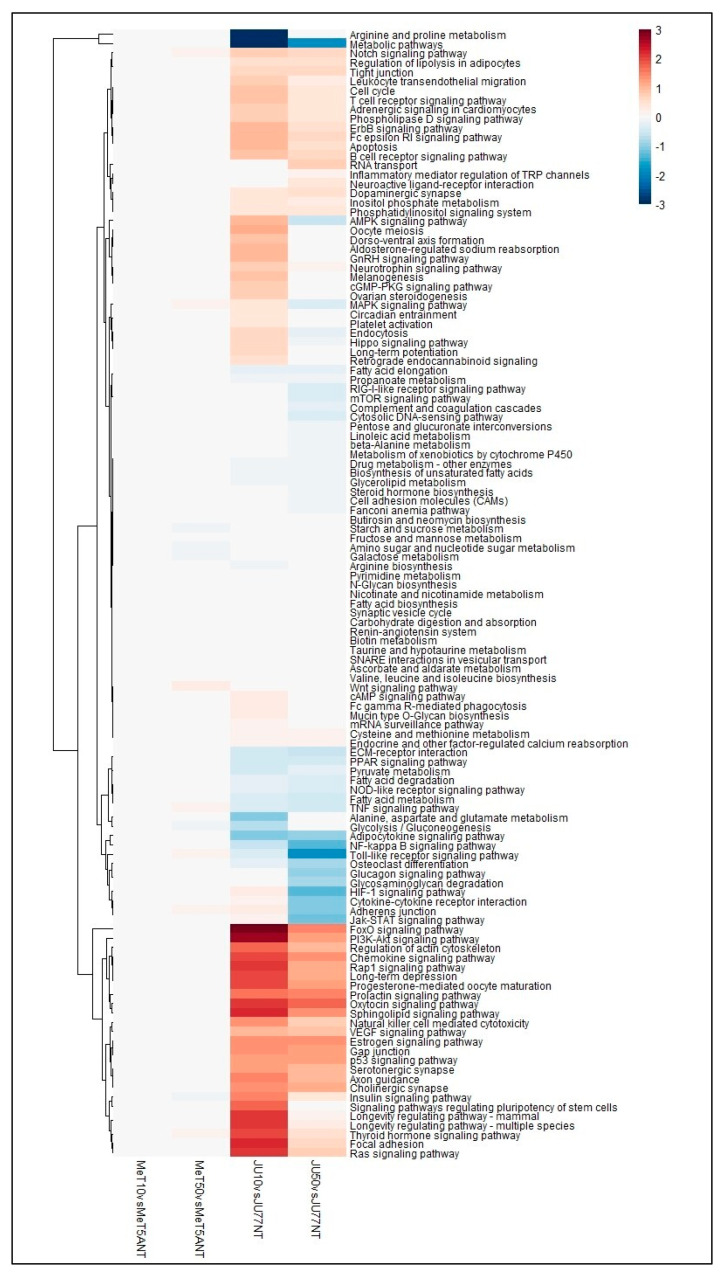
Heatmap showing the dysregulated pathways between untreated vs. FE fibers treated JU77 (human malignant mesothelioma) and MeT-5A (human normal mesothelium) by taking into account the differentially expressed miRNAs.

**Table 1 biomedicines-11-00538-t001:** Interspecies differentially expressed miRNAs.

JU77NTvsMeT5ANT		JU10vsMeT10		JU50vsMeT50	
Up-Regulated miRNAs	Down-Regulated miRNAs	Up-Regulated miRNAs	Down-Regulated miRNAs	Up-Regulated miRNAs	Down-Regulated miRNAs
hsa-miR-144-5p	hsa-miR-196b-3p	hsa-miR-139-5p	hsa-miR-4466	hsa-miR-139-5p	hsa-miR-23a-5p
hsa-miR-139-5p	hsa-miR-550a-3p	hsa-miR-328-3p	hsa-miR-550a-3p	hsa-miR-328-3p	hsa-miR-615-3p
hsa-miR-328-3p	hsa-miR-615-3p	hsa-miR-549a	hsa-miR-615-3p	hsa-miR-549a	hsa-miR-3912-5p
hsa-miR-3661	hsa-miR-3912-5p	hsa-miR-3167	hsa-miR-3912-5p	hsa-miR-3167	hsa-miR-2116-5p
hsa-miR-549a	hsa-miR-2116-5p	hsa-miR-6844	hsa-miR-2116-5p	hsa-miR-6844	hsa-miR-3618
hsa-miR-3140-5p	hsa-miR-3613-3p	hsa-miR-18b-5p	hsa-miR-3613-3p	hsa-miR-18b-5p	hsa-miR-1262
hsa-miR-3167	hsa-miR-545-3p	hsa-let-7a-2-3p	hsa-miR-627-3p	hsa-let-7a-2-3p	hsa-miR-143-3p
hsa-miR-6844	hsa-miR-627-3p	hsa-let-7c-3p	hsa-miR-664a-5p	hsa-let-7c-3p	hsa-miR-16-1-3p
hsa-miR-18b-5p	hsa-miR-664a-5p	hsa-miR-100-3p	hsa-miR-3618	hsa-miR-100-3p	hsa-miR-215-5p
hsa-miR-4466	hsa-miR-935	hsa-miR-125b-2-3p	hsa-miR-1262	hsa-miR-125b-2-3p	hsa-miR-25-5p
hsa-let-7a-2-3p	hsa-miR-3618	hsa-miR-222-5p	hsa-miR-143-3p	hsa-miR-222-5p	hsa-miR-3176
hsa-let-7c-3p	hsa-miR-1262	hsa-miR-29a-5p	hsa-miR-16-1-3p	hsa-miR-29a-5p	hsa-miR-4326
hsa-miR-100-3p	hsa-miR-143-3p	hsa-miR-29b-1-5p	hsa-miR-215-5p	hsa-miR-29b-1-5p	hsa-miR-484
hsa-miR-125b-2-3p	hsa-miR-16-1-3p	hsa-miR-363-3p	hsa-miR-25-5p	hsa-miR-363-3p	hsa-miR-7-5p_1
hsa-miR-128-1-5p	hsa-miR-215-5p	hsa-miR-449c-5p	hsa-miR-3176	hsa-miR-449c-5p	hsa-let-7f-2-3p
hsa-miR-152-5p	hsa-miR-25-5p	hsa-miR-501-5p	hsa-miR-4326	hsa-miR-501-5p	hsa-miR-16-2-3p
hsa-miR-19a-5p	hsa-miR-3176	hsa-miR-503-3p	hsa-miR-484	hsa-miR-503-3p	hsa-miR-2278
hsa-miR-200a-3p	hsa-miR-4326	hsa-miR-410-3p	hsa-miR-7-5p_1	hsa-miR-410-3p	hsa-miR-30d-3p
hsa-miR-200b-3p	hsa-miR-484	hsa-miR-125b-1-3p	hsa-let-7f-2-3p	hsa-miR-125b-1-3p	hsa-miR-3129-5p
hsa-miR-222-5p	hsa-miR-7-5p_1	hsa-miR-1272	hsa-miR-16-2-3p	hsa-miR-1272	hsa-miR-340-3p
hsa-miR-29a-5p	hsa-let-7f-2-3p	hsa-miR-188-5p	hsa-miR-2278	hsa-miR-433-3p	hsa-miR-34c-5p
hsa-miR-29b-1-5p	hsa-miR-16-2-3p	hsa-miR-23a-5p	hsa-miR-30d-3p	hsa-miR-548u	hsa-miR-454-5p
hsa-miR-362-5p	hsa-miR-2278	hsa-miR-3934-5p	hsa-miR-3129-5p	hsa-miR-99a-5p	hsa-miR-561-3p
hsa-miR-363-3p	hsa-miR-30d-3p	hsa-miR-433-3p	hsa-miR-340-3p	hsa-miR-7706	hsa-miR-570-3p
hsa-miR-429	hsa-miR-3129-5p	hsa-miR-495-5p	hsa-miR-34c-5p	hsa-miR-135b-5p	hsa-miR-579-3p
hsa-miR-449c-5p	hsa-miR-340-3p	hsa-miR-548u	hsa-miR-454-5p	hsa-miR-154-3p	hsa-miR-624-3p
hsa-miR-4517	hsa-miR-34c-5p	hsa-miR-99a-5p	hsa-miR-561-3p	hsa-miR-221-5p	hsa-miR-624-5p
hsa-miR-4521	hsa-miR-454-5p	hsa-miR-7706	hsa-miR-570-3p	hsa-miR-660-3p	hsa-miR-627-5p
hsa-miR-501-5p	hsa-miR-561-3p	hsa-miR-135b-5p	hsa-miR-579-3p	hsa-miR-496	hsa-miR-7-1-3p
hsa-miR-503-3p	hsa-miR-570-3p	hsa-miR-449a	hsa-miR-624-3p	hsa-miR-636	hsa-miR-3653
hsa-miR-573	hsa-miR-579-3p	hsa-miR-154-3p	hsa-miR-624-5p	hsa-miR-6720-3p	hsa-miR-1278
hsa-miR-6513-3p	hsa-miR-624-3p	hsa-miR-221-5p	hsa-miR-627-5p	hsa-miR-5586-5p	hsa-miR-195-3p
hsa-miR-410-3p	hsa-miR-624-5p	hsa-miR-660-3p	hsa-miR-7-1-3p	hsa-let-7c-5p	hsa-miR-23b-5p
hsa-miR-125b-1-3p	hsa-miR-627-5p	hsa-miR-496	hsa-miR-3653	hsa-miR-146a-5p	hsa-miR-33b-5p
hsa-miR-1272	hsa-miR-7-1-3p	hsa-miR-636	hsa-miR-1278	hsa-miR-708-5p	hsa-miR-548j-5p
hsa-miR-128-3p_1	hsa-miR-3653	hsa-miR-6720-3p	hsa-miR-195-3p	hsa-miR-431-3p	hsa-miR-7976
hsa-miR-188-5p	hsa-miR-4645-3p	hsa-miR-5586-5p	hsa-miR-23b-5p	hsa-miR-1226-3p	hsa-miR-664a-3p
hsa-miR-23a-5p	hsa-miR-1278	hsa-let-7c-5p	hsa-miR-33b-5p	hsa-miR-218-1-3p	hsa-let-7f-1-3p
hsa-miR-3934-5p	hsa-miR-195-3p	hsa-miR-146a-5p	hsa-miR-548j-5p	hsa-let-7a-3p	hsa-miR-106b-3p
hsa-miR-433-3p	hsa-miR-23b-5p	hsa-miR-708-5p	hsa-miR-7976	hsa-let-7a-5p	hsa-miR-132-3p
hsa-miR-487b-5p	hsa-miR-33b-3p	hsa-miR-431-3p	hsa-miR-1248	hsa-let-7a-5p_1	hsa-miR-181b-3p
hsa-miR-495-5p	hsa-miR-33b-5p	hsa-miR-629-3p	hsa-miR-3651	hsa-miR-100-5p	hsa-miR-3173-5p
hsa-miR-548u	hsa-miR-380-3p	hsa-miR-1226-3p	hsa-miR-664a-3p	hsa-miR-106a-5p	hsa-miR-345-5p
hsa-miR-6747-3p	hsa-miR-380-5p	hsa-miR-218-1-3p	hsa-let-7f-1-3p	hsa-miR-10a-3p	hsa-miR-3611
hsa-miR-99a-5p	hsa-miR-4746-5p	hsa-let-7a-3p	hsa-miR-106b-3p	hsa-miR-10a-5p	hsa-miR-92a-3p
hsa-miR-7706	hsa-miR-548j-5p	hsa-let-7a-5p	hsa-miR-132-3p	hsa-miR-10b-5p	hsa-miR-93-3p
hsa-miR-135b-5p	hsa-miR-7976	hsa-let-7a-5p_1	hsa-miR-181b-3p	hsa-miR-125b-5p	hsa-miR-1291
hsa-miR-92a-1-5p	hsa-miR-1248	hsa-miR-100-5p	hsa-miR-3173-5p	hsa-miR-125b-5p_1	hsa-miR-3614-3p
hsa-miR-6514-5p	hsa-miR-3651	hsa-miR-106a-5p	hsa-miR-345-5p	hsa-miR-137	hsa-miR-615-5p
hsa-miR-449a	hsa-miR-664a-3p	hsa-miR-10a-3p	hsa-miR-3611	hsa-miR-138-5p_1	hsa-let-7a-5p_2
hsa-miR-154-3p	hsa-let-7f-1-3p	hsa-miR-10a-5p	hsa-miR-92a-3p	hsa-miR-148b-3p	hsa-let-7d-3p
hsa-miR-17-5p	hsa-miR-106b-3p	hsa-miR-10b-5p	hsa-miR-93-3p	hsa-miR-155-5p	hsa-let-7d-5p
hsa-miR-190b	hsa-miR-132-3p	hsa-miR-125b-5p	hsa-miR-1291	hsa-miR-193a-3p	hsa-let-7f-5p
hsa-miR-221-5p	hsa-miR-181b-3p	hsa-miR-125b-5p_1	hsa-miR-3614-3p	hsa-miR-193a-5p	hsa-let-7f-5p_1
hsa-miR-424-3p	hsa-miR-3173-5p	hsa-miR-137	hsa-miR-615-5p	hsa-miR-20b-5p	hsa-let-7i-3p
hsa-miR-487a-3p	hsa-miR-345-5p	hsa-miR-138-5p_1	hsa-let-7a-5p_2	hsa-miR-218-5p	hsa-let-7i-5p
hsa-miR-502-5p	hsa-miR-3611	hsa-miR-148b-3p	hsa-let-7d-3p	hsa-miR-218-5p_1	hsa-miR-101-3p
hsa-miR-660-3p	hsa-miR-92a-3p	hsa-miR-155-5p	hsa-let-7d-5p	hsa-miR-221-3p	hsa-miR-106b-5p
hsa-miR-487a-5p	hsa-miR-93-3p	hsa-miR-193a-3p	hsa-let-7f-5p	hsa-miR-222-3p	hsa-miR-107
hsa-miR-496	hsa-miR-1291	hsa-miR-193a-5p	hsa-let-7f-5p_1	hsa-miR-27a-5p	hsa-miR-1179
hsa-miR-636	hsa-miR-6803-3p	hsa-miR-20b-5p	hsa-let-7i-3p	hsa-miR-29a-3p	hsa-miR-1180-3p
hsa-miR-6720-3p	hsa-miR-2110	hsa-miR-218-5p	hsa-let-7i-5p	hsa-miR-29b-3p_1	hsa-miR-125a-5p
hsa-miR-5586-5p	hsa-miR-3614-3p	hsa-miR-218-5p_1	hsa-miR-101-3p	hsa-miR-30c-5p_1	hsa-miR-126-3p
hsa-let-7c-5p	hsa-miR-381-5p	hsa-miR-221-3p	hsa-miR-106b-5p	hsa-miR-3188	hsa-miR-126-5p
hsa-miR-146a-5p	hsa-miR-615-5p	hsa-miR-222-3p	hsa-miR-107	hsa-miR-326	hsa-miR-1271-5p
hsa-miR-708-5p	hsa-let-7a-5p_2	hsa-miR-27a-5p	hsa-miR-1179	hsa-miR-335-3p	hsa-miR-128-3p
hsa-miR-1228-3p	hsa-let-7b-5p	hsa-miR-29a-3p	hsa-miR-1180-3p	hsa-miR-34a-3p	hsa-miR-1296-5p
hsa-miR-370-5p	hsa-let-7d-3p	hsa-miR-29b-3p_1	hsa-miR-125a-5p	hsa-miR-34a-5p	hsa-miR-130b-3p
hsa-miR-431-3p	hsa-let-7d-5p	hsa-miR-30c-5p_1	hsa-miR-126-3p	hsa-miR-412-5p	hsa-miR-130b-5p
hsa-miR-501-3p	hsa-let-7f-5p	hsa-miR-3188	hsa-miR-126-5p	hsa-miR-431-5p	hsa-miR-142-3p
hsa-miR-629-3p	hsa-let-7f-5p_1	hsa-miR-326	hsa-miR-1271-5p	hsa-miR-432-5p	hsa-miR-146b-5p
hsa-miR-1226-3p	hsa-let-7i-3p	hsa-miR-335-3p	hsa-miR-128-3p	hsa-miR-455-3p	hsa-miR-148a-3p
hsa-miR-218-1-3p	hsa-let-7i-5p	hsa-miR-34a-3p	hsa-miR-1296-5p	hsa-miR-455-5p	hsa-miR-148a-5p
hsa-let-7a-3p	hsa-miR-101-3p	hsa-miR-34a-5p	hsa-miR-1301-3p	hsa-miR-495-3p	hsa-miR-149-5p
hsa-let-7a-5p	hsa-miR-106b-5p	hsa-miR-376b-3p	hsa-miR-130b-3p	hsa-miR-5001-3p	hsa-miR-151a-5p
hsa-let-7a-5p_1	hsa-miR-107	hsa-miR-412-5p	hsa-miR-130b-5p	hsa-miR-543	hsa-miR-15a-5p
hsa-miR-100-5p	hsa-miR-1179	hsa-miR-431-5p	hsa-miR-142-3p	hsa-miR-574-3p	hsa-miR-15b-3p
hsa-miR-106a-5p	hsa-miR-1180-3p	hsa-miR-432-5p	hsa-miR-146b-5p	hsa-miR-584-5p	hsa-miR-15b-5p
hsa-miR-10a-3p	hsa-miR-125a-5p	hsa-miR-455-3p	hsa-miR-148a-3p	hsa-miR-625-5p	hsa-miR-16-5p
hsa-miR-10a-5p	hsa-miR-126-3p	hsa-miR-455-5p	hsa-miR-148a-5p	hsa-miR-628-3p	hsa-miR-16-5p_1
hsa-miR-10b-5p	hsa-miR-126-5p	hsa-miR-495-3p	hsa-miR-149-5p	hsa-miR-628-5p	hsa-miR-181a-3p
hsa-miR-1185-1-3p	hsa-miR-1271-5p	hsa-miR-5001-3p	hsa-miR-151a-5p	hsa-miR-660-5p	hsa-miR-181a-5p
hsa-miR-1185-2-3p	hsa-miR-128-3p	hsa-miR-543	hsa-miR-15a-5p	hsa-miR-6720-5p	hsa-miR-181a-5p_1
hsa-miR-125b-5p	hsa-miR-1296-5p	hsa-miR-574-3p	hsa-miR-15b-3p	hsa-miR-99a-3p	hsa-miR-181b-5p_1
hsa-miR-125b-5p_1	hsa-miR-1301-3p	hsa-miR-584-5p	hsa-miR-15b-5p	hsa-miR-4797-3p	hsa-miR-182-5p
hsa-miR-137	hsa-miR-130b-3p	hsa-miR-625-5p	hsa-miR-16-5p	hsa-miR-6783-5p	hsa-miR-185-3p
hsa-miR-138-5p_1	hsa-miR-130b-5p	hsa-miR-628-3p	hsa-miR-16-5p_1	hsa-miR-101-5p	hsa-miR-185-5p
hsa-miR-148b-3p	hsa-miR-134-5p	hsa-miR-628-5p	hsa-miR-181a-3p	hsa-miR-548aw	hsa-miR-191-5p
hsa-miR-148b-5p	hsa-miR-142-3p	hsa-miR-660-5p	hsa-miR-181a-5p	hsa-miR-335-5p	hsa-miR-192-5p
hsa-miR-155-5p	hsa-miR-146b-5p	hsa-miR-6720-5p	hsa-miR-181a-5p_1	hsa-miR-4742-3p	hsa-miR-193b-3p
hsa-miR-193a-3p	hsa-miR-148a-3p	hsa-miR-99a-3p	hsa-miR-181b-5p_1	hsa-miR-3138	hsa-miR-195-5p
hsa-miR-193a-5p	hsa-miR-148a-5p	hsa-miR-4797-3p	hsa-miR-182-5p	hsa-miR-877-5p	hsa-miR-196b-5p
hsa-miR-20b-5p	hsa-miR-149-5p	hsa-miR-6783-5p	hsa-miR-185-3p	hsa-miR-144-5p	hsa-miR-19a-3p
hsa-miR-218-5p	hsa-miR-151a-5p	hsa-miR-101-5p	hsa-miR-185-5p	hsa-miR-3661	hsa-miR-210-3p
hsa-miR-218-5p_1	hsa-miR-15a-5p	hsa-miR-199b-5p	hsa-miR-191-5p	hsa-miR-429	hsa-miR-2116-3p
hsa-miR-221-3p	hsa-miR-15b-3p	hsa-miR-548aw	hsa-miR-192-5p	hsa-miR-4521	hsa-miR-21-3p
hsa-miR-222-3p	hsa-miR-15b-5p	hsa-miR-6716-3p	hsa-miR-193b-3p	hsa-miR-573	hsa-miR-21-5p
hsa-miR-2277-5p	hsa-miR-16-5p	hsa-miR-449b-5p	hsa-miR-195-5p	hsa-miR-6513-3p	hsa-miR-219a-5p
hsa-miR-27a-3p	hsa-miR-16-5p_1	hsa-miR-130a-5p	hsa-miR-196b-5p	hsa-miR-92a-1-5p	hsa-miR-224-5p
hsa-miR-27a-5p	hsa-miR-181a-3p	hsa-miR-335-5p	hsa-miR-19a-3p	hsa-miR-6514-5p	hsa-miR-22-5p
hsa-miR-29a-3p	hsa-miR-181a-5p	hsa-miR-551a	hsa-miR-210-3p	hsa-miR-190b	hsa-miR-2355-3p
hsa-miR-29b-2-5p	hsa-miR-181a-5p_1		hsa-miR-2116-3p	hsa-miR-502-5p	hsa-miR-2355-5p
hsa-miR-29b-3p	hsa-miR-181b-5p_1		hsa-miR-21-3p	hsa-miR-487a-5p	hsa-miR-23b-3p
hsa-miR-29b-3p_1	hsa-miR-182-5p		hsa-miR-21-5p	hsa-miR-1228-3p	hsa-miR-25-3p
hsa-miR-30c-2-3p	hsa-miR-185-3p		hsa-miR-219a-5p	hsa-miR-501-3p	hsa-miR-27b-3p
hsa-miR-30c-5p_1	hsa-miR-185-5p		hsa-miR-224-5p	hsa-miR-2277-5p	hsa-miR-27b-5p
hsa-miR-31-5p	hsa-miR-191-5p		hsa-miR-22-5p	hsa-miR-29b-2-5p	hsa-miR-28-3p
hsa-miR-3188	hsa-miR-192-5p		hsa-miR-2355-3p	hsa-miR-30c-2-3p	hsa-miR-28-5p
hsa-miR-326	hsa-miR-193b-3p		hsa-miR-2355-5p	hsa-miR-409-5p	hsa-miR-299-3p
hsa-miR-335-3p	hsa-miR-195-5p		hsa-miR-23b-3p	hsa-miR-485-3p	hsa-miR-299-5p
hsa-miR-339-5p	hsa-miR-196a-5p_1		hsa-miR-25-3p	hsa-miR-493-3p	hsa-miR-301b
hsa-miR-34a-3p	hsa-miR-196b-5p		hsa-miR-26a-5p_1	hsa-miR-500a-3p	hsa-miR-3074-3p
hsa-miR-34a-5p	hsa-miR-19a-3p		hsa-miR-27b-3p	hsa-miR-502-3p	hsa-miR-30b-5p
hsa-miR-3605-3p	hsa-miR-210-3p		hsa-miR-27b-5p	hsa-miR-505-5p	hsa-miR-30d-5p
hsa-miR-376a-2-5p	hsa-miR-2116-3p		hsa-miR-28-3p	hsa-miR-532-3p	hsa-miR-3129-3p
hsa-miR-376a-5p	hsa-miR-212-3p		hsa-miR-28-5p	hsa-miR-532-5p	hsa-miR-3200-3p
hsa-miR-376b-3p	hsa-miR-21-3p		hsa-miR-299-3p	hsa-miR-92a-3p_1	hsa-miR-32-5p
hsa-miR-377-3p	hsa-miR-21-5p		hsa-miR-299-5p	hsa-miR-933	hsa-miR-33a-3p
hsa-miR-409-5p	hsa-miR-219a-5p		hsa-miR-301b	hsa-miR-370-3p	hsa-miR-33a-5p
hsa-miR-412-5p	hsa-miR-224-5p		hsa-miR-3074-3p	hsa-miR-181c-3p	hsa-miR-340-5p
hsa-miR-431-5p	hsa-miR-22-5p		hsa-miR-30b-5p	hsa-miR-183-3p	hsa-miR-342-3p
hsa-miR-432-5p	hsa-miR-2355-3p		hsa-miR-30d-5p	hsa-miR-337-3p	hsa-miR-342-5p
hsa-miR-455-3p	hsa-miR-2355-5p		hsa-miR-3129-3p	hsa-miR-4677-5p	hsa-miR-34b-5p
hsa-miR-455-5p	hsa-miR-23b-3p		hsa-miR-3136-5p	hsa-miR-625-3p	hsa-miR-3613-5p
hsa-miR-4775	hsa-miR-25-3p		hsa-miR-3200-3p		hsa-miR-361-3p
hsa-miR-485-3p	hsa-miR-26a-5p_1		hsa-miR-324-5p		hsa-miR-361-5p
hsa-miR-485-5p	hsa-miR-27b-3p		hsa-miR-32-5p		hsa-miR-374a-3p
hsa-miR-493-3p	hsa-miR-27b-5p		hsa-miR-33a-3p		hsa-miR-374a-5p
hsa-miR-495-3p	hsa-miR-28-3p		hsa-miR-33a-5p		hsa-miR-376b-5p
hsa-miR-5001-3p	hsa-miR-28-5p		hsa-miR-340-5p		hsa-miR-3912-3p
hsa-miR-500a-3p	hsa-miR-299-3p		hsa-miR-342-3p		hsa-miR-450a-1-3p
hsa-miR-502-3p	hsa-miR-299-5p		hsa-miR-342-5p		hsa-miR-450a-2-3p
hsa-miR-505-5p	hsa-miR-301b		hsa-miR-34b-5p		hsa-miR-450b-5p
hsa-miR-532-3p	hsa-miR-3074-3p		hsa-miR-3613-5p		hsa-miR-452-3p
hsa-miR-532-5p	hsa-miR-30b-5p		hsa-miR-361-3p		hsa-miR-452-5p
hsa-miR-543	hsa-miR-30d-5p		hsa-miR-361-5p		hsa-miR-454-3p
hsa-miR-574-3p	hsa-miR-3129-3p		hsa-miR-374a-3p		hsa-miR-4766-3p
hsa-miR-584-5p	hsa-miR-3136-5p		hsa-miR-374a-5p		hsa-miR-491-5p
hsa-miR-625-5p	hsa-miR-3200-3p		hsa-miR-376b-5p		hsa-miR-494-3p
hsa-miR-628-3p	hsa-miR-324-5p		hsa-miR-3912-3p		hsa-miR-497-5p
hsa-miR-628-5p	hsa-miR-32-5p		hsa-miR-450a-1-3p		hsa-miR-5008-5p
hsa-miR-660-5p	hsa-miR-33a-3p		hsa-miR-450a-2-3p		hsa-miR-542-3p
hsa-miR-6720-5p	hsa-miR-33a-5p		hsa-miR-450b-5p		hsa-miR-542-5p
hsa-miR-92a-3p_1	hsa-miR-340-5p		hsa-miR-452-3p		hsa-miR-556-3p
hsa-miR-99a-3p	hsa-miR-342-3p		hsa-miR-452-5p		hsa-miR-561-5p
	hsa-miR-342-5p		hsa-miR-454-3p		hsa-miR-5699-3p
	hsa-miR-34b-5p		hsa-miR-4766-3p		hsa-miR-5699-5p
	hsa-miR-3613-5p		hsa-miR-491-5p		hsa-miR-585-3p
	hsa-miR-361-3p		hsa-miR-494-3p		hsa-miR-590-3p
	hsa-miR-361-5p		hsa-miR-497-5p		hsa-miR-590-5p
	hsa-miR-374a-3p		hsa-miR-5008-5p		hsa-miR-598-3p
	hsa-miR-374a-5p		hsa-miR-542-3p		hsa-miR-671-5p
	hsa-miR-376b-5p		hsa-miR-542-5p		hsa-miR-7705
	hsa-miR-3912-3p		hsa-miR-556-3p		hsa-miR-92b-3p
	hsa-miR-450a-1-3p		hsa-miR-561-5p		hsa-miR-92b-5p
	hsa-miR-450a-2-3p		hsa-miR-5699-3p		hsa-miR-93-5p
	hsa-miR-450b-5p		hsa-miR-5699-5p		hsa-miR-942-5p
	hsa-miR-452-3p		hsa-miR-585-3p		hsa-miR-95-3p
	hsa-miR-452-5p		hsa-miR-590-3p		hsa-miR-96-5p
	hsa-miR-454-3p		hsa-miR-590-5p		hsa-miR-99b-3p
	hsa-miR-4766-3p		hsa-miR-598-3p		hsa-miR-579-5p
	hsa-miR-491-5p		hsa-miR-641		hsa-miR-3657
	hsa-miR-494-3p		hsa-miR-671-5p		hsa-let-7g-3p
	hsa-miR-497-5p		hsa-miR-7705		hsa-miR-1305
	hsa-miR-5008-5p		hsa-miR-92b-3p		hsa-miR-132-5p
	hsa-miR-542-3p		hsa-miR-92b-5p		hsa-miR-15a-3p
	hsa-miR-542-5p		hsa-miR-93-5p		hsa-miR-32-3p
	hsa-miR-556-3p		hsa-miR-942-5p		hsa-miR-548k
	hsa-miR-561-5p		hsa-miR-95-3p		hsa-miR-7974
	hsa-miR-5699-3p		hsa-miR-96-5p		hsa-miR-19b-3p_1
	hsa-miR-5699-5p		hsa-miR-99b-3p		hsa-miR-30b-3p
	hsa-miR-585-3p		hsa-miR-186-3p		hsa-miR-580-3p
	hsa-miR-590-3p		hsa-miR-579-5p		hsa-miR-671-3p
	hsa-miR-590-5p		hsa-miR-3657		hsa-miR-1197
	hsa-miR-598-3p		hsa-let-7g-3p		hsa-miR-140-3p
	hsa-miR-641		hsa-miR-1305		hsa-miR-192-3p
	hsa-miR-651-5p		hsa-miR-132-5p		hsa-miR-219a-1-3p
	hsa-miR-655-3p		hsa-miR-15a-3p		hsa-miR-22-3p
	hsa-miR-671-5p		hsa-miR-32-3p		hsa-miR-296-5p
	hsa-miR-7705		hsa-miR-548k		hsa-miR-323a-3p
	hsa-miR-92b-3p		hsa-miR-7974		hsa-miR-378a-5p
	hsa-miR-92b-5p		hsa-miR-19b-3p_1		hsa-miR-381-3p
	hsa-miR-93-5p		hsa-miR-30b-3p		hsa-miR-5582-3p
	hsa-miR-942-5p		hsa-miR-580-3p		hsa-miR-643
	hsa-miR-95-3p		hsa-miR-671-3p		hsa-miR-19a-5p
	hsa-miR-96-5p		hsa-miR-4742-3p		hsa-miR-196b-3p
	hsa-miR-99b-3p		hsa-miR-1296-3p		hsa-miR-4645-3p
	hsa-miR-99b-5p		hsa-miR-26a-1-3p		hsa-miR-33b-3p
			hsa-miR-3187-3p		hsa-miR-380-3p
			hsa-miR-3684		hsa-miR-4746-5p
			hsa-miR-548e-3p		hsa-miR-3140-3p
			hsa-miR-744-3p		hsa-miR-19b-1-5p
			hsa-miR-200c-3p		hsa-miR-362-3p
			hsa-miR-3138		hsa-miR-320b_1
			hsa-miR-330-3p		hsa-miR-331-5p
			hsa-miR-877-5p		hsa-miR-548n
			hsa-let-7g-5p		hsa-miR-651-3p
			hsa-miR-1197		hsa-miR-758-5p
			hsa-miR-125a-3p		hsa-miR-193b-5p
			hsa-miR-1292-5p		hsa-miR-323b-3p
			hsa-miR-1307-5p		hsa-miR-548e-5p
			hsa-miR-140-3p		
			hsa-miR-191-3p		
			hsa-miR-192-3p		
			hsa-miR-194-3p		
			hsa-miR-197-3p		
			hsa-miR-219a-1-3p		
			hsa-miR-22-3p		
			hsa-miR-26a-5p		
			hsa-miR-296-5p		
			hsa-miR-323a-3p		
			hsa-miR-324-3p		
			hsa-miR-330-5p		
			hsa-miR-3615		
			hsa-miR-3679-5p		
			hsa-miR-378a-5p		
			hsa-miR-381-3p		
			hsa-miR-3940-3p		
			hsa-miR-5582-3p		
			hsa-miR-643		
			hsa-miR-656-3p		
			hsa-miR-744-5p		

**Table 2 biomedicines-11-00538-t002:** Interspecies differentially expressed miRNAs in common.

**Up-Regulated miRNAs in Common**		
**JU77NTvsMeT5ANT and JU10vsMeT10**	**JU77NTvsMeT5ANT and JU50vsMeT50**	**JU10vsMeT10 and JU50vsMeT50**
hsa-miR-376b-3p	hsa-miR-4521	hsa-miR-335-5p
hsa-miR-188-5p	hsa-miR-92a-3p_1	hsa-miR-5586-5p
hsa-miR-410-3p	hsa-miR-532-5p	
hsa-miR-29b-3p_1	hsa-miR-429	
hsa-miR-449a	hsa-miR-485-3p	
hsa-miR-548u	hsa-miR-30c-2-3p	
	hsa-miR-493-3p	
	hsa-miR-29b-2-5p	
	hsa-miR-190b	
	hsa-miR-502-3p	
	hsa-miR-532-3p	
	hsa-miR-500a-3p	
	hsa-miR-409-5p	
	hsa-miR-431-3p	
	hsa-miR-660-3p	
	hsa-miR-505-5p	
	hsa-miR-502-5p	
	hsa-miR-2277-5p	
**Down-Regulated miRNAs in Common**		
**JU77NTvsMeT5ANT and JU10vsMeT10**	**JU77NTvsMeT5ANT and JU50vsMeT50**	**JU10vsMeT10 and JU50vsMeT50**
hsa-miR-1248	hsa-miR-4645-3p	hsa-miR-22-3p
hsa-miR-26a-5p_1	hsa-miR-33a-3p	hsa-miR-140-3p
hsa-miR-324-5p	hsa-miR-2355-5p	hsa-miR-381-3p
hsa-miR-1301-3p		hsa-miR-132-5p
hsa-miR-664a-3p		hsa-miR-3176
hsa-miR-641		hsa-miR-323a-3p
hsa-miR-3651		hsa-miR-296-5p
hsa-miR-627-3p		hsa-let-7g-3p
		hsa-miR-19b-3p_1
		hsa-miR-32-3p
		hsa-miR-378a-5p
		hsa-miR-15a-3p
		hsa-miR-548k
		hsa-miR-7974
		hsa-miR-671-3p
		hsa-miR-561-3p
		hsa-miR-7-5p_1

**Table 3 biomedicines-11-00538-t003:** Intraspecies differentially expressed microRNAs.

MeT10vsMeT5ANT		MeT50vsMeT5ANT		JU10vsJU77NT		JU50vsJU77NT	
Up-Regulated miRNAs	Down-Regulated miRNAs	Up-Regulated miRNAs	Down-Regulated miRNAs	Up-Regulated miRNAs	Down-Regulated miRNAs	Up-Regulated miRNAs	Down-Regulated miRNAs
hsa-miR-1228-3p	hsa-miR-1226-3p	hsa-miR-144-5p	hsa-miR-1248	hsa-miR-3653	hsa-miR-3140-5p	hsa-miR-4797-3p	hsa-miR-3140-3p
hsa-miR-144-5p	hsa-miR-1248	hsa-miR-3140-5p	hsa-miR-1291	hsa-miR-4645-3p	hsa-miR-3167	hsa-miR-6783-5p	hsa-miR-3140-5p
hsa-miR-18b-5p	hsa-miR-1291	hsa-miR-3167	hsa-miR-3651	hsa-miR-539-5p	hsa-miR-1248	hsa-miR-101-5p	hsa-miR-3167
hsa-miR-193b-5p	hsa-miR-139-5p	hsa-miR-487a-5p	hsa-miR-3912-5p	hsa-miR-3157-5p	hsa-miR-3651	hsa-miR-196b-3p	hsa-miR-186-3p
hsa-miR-3140-5p	hsa-miR-218-1-3p	hsa-miR-496	hsa-miR-548l	hsa-miR-4797-3p	hsa-miR-135b-5p	hsa-miR-199b-5p	hsa-miR-3618
hsa-miR-3157-5p	hsa-miR-3614-3p	hsa-miR-636	hsa-miR-5586-5p	hsa-miR-6783-5p	hsa-miR-186-3p	hsa-miR-548aw	hsa-miR-579-5p
hsa-miR-3167	hsa-miR-3651	hsa-miR-6720-3p	hsa-miR-652-3p	hsa-miR-7706	hsa-miR-3618	hsa-miR-550a-3p	hsa-miR-6844
hsa-miR-330-3p	hsa-miR-3657	hsa-miR-758-5p	hsa-miR-6803-3p	hsa-miR-101-5p	hsa-miR-4742-3p	hsa-miR-615-3p	hsa-miR-18b-5p
hsa-miR-370-5p	hsa-miR-381-5p		hsa-let-7c-5p	hsa-miR-1278	hsa-miR-579-5p	hsa-miR-144-5p	hsa-miR-4466
hsa-miR-431-3p	hsa-miR-3912-5p		hsa-miR-135b-5p	hsa-miR-195-3p	hsa-miR-664a-3p	hsa-miR-3912-5p	hsa-miR-3657
hsa-miR-4466	hsa-miR-410-3p		hsa-miR-146a-5p	hsa-miR-196b-3p	hsa-miR-6844	hsa-miR-933	hsa-miR-6716-3p
hsa-miR-4797-3p	hsa-miR-449a		hsa-miR-186-3p	hsa-miR-199b-5p	hsa-miR-92a-1-5p	hsa-miR-139-5p	hsa-let-7a-2-3p
hsa-miR-487a-5p	hsa-miR-548l		hsa-miR-200c-3p	hsa-miR-23b-5p	hsa-miR-18b-5p	hsa-miR-2116-5p	hsa-let-7c-3p
hsa-miR-496	hsa-miR-5586-5p		hsa-miR-2110	hsa-miR-3140-3p	hsa-miR-4466	hsa-miR-328-3p	hsa-let-7g-3p
hsa-miR-501-3p	hsa-miR-615-5p		hsa-miR-3138	hsa-miR-33b-3p	hsa-miR-6514-5p	hsa-miR-3613-3p	hsa-miR-100-3p
hsa-miR-629-3p	hsa-miR-652-3p		hsa-miR-3143	hsa-miR-33b-5p	hsa-miR-3657	hsa-miR-3661	hsa-miR-125b-2-3p
hsa-miR-636	hsa-miR-6716-3p		hsa-miR-3618	hsa-miR-380-3p	hsa-miR-449a	hsa-miR-370-3p	hsa-miR-1262
hsa-miR-6514-5p	hsa-miR-6803-3p		hsa-miR-3653	hsa-miR-380-5p	hsa-miR-6716-3p	hsa-miR-545-3p	hsa-miR-128-1-5p
hsa-miR-6783-5p			hsa-miR-3934-3p	hsa-miR-4746-5p	hsa-let-7a-2-3p	hsa-miR-549a	hsa-miR-1305
hsa-miR-7706			hsa-miR-4645-3p	hsa-miR-548aw	hsa-let-7c-3p	hsa-miR-627-3p	hsa-miR-132-5p
hsa-miR-877-5p			hsa-miR-4742-3p	hsa-miR-548j-5p	hsa-let-7f-1-3p	hsa-miR-664a-5p	hsa-miR-143-3p
hsa-miR-933			hsa-miR-539-5p	hsa-miR-550a-3p	hsa-let-7g-3p	hsa-miR-935	hsa-miR-152-5p
			hsa-miR-579-5p	hsa-miR-615-3p	hsa-miR-100-3p		hsa-miR-15a-3p
			hsa-miR-664a-3p	hsa-miR-7976	hsa-miR-106b-3p		hsa-miR-16-1-3p
			hsa-miR-6844		hsa-miR-125b-2-3p		hsa-miR-18a-3p
			hsa-miR-708-5p		hsa-miR-1262		hsa-miR-18a-5p
			hsa-miR-92a-1-5p		hsa-miR-128-1-5p		hsa-miR-19a-5p
					hsa-miR-1296-3p		hsa-miR-19b-1-5p
					hsa-miR-1305		hsa-miR-200a-3p
					hsa-miR-132-3p		hsa-miR-200b-3p
					hsa-miR-132-5p		hsa-miR-20a-3p
					hsa-miR-136-5p		hsa-miR-215-5p
					hsa-miR-143-3p		hsa-miR-222-5p
					hsa-miR-152-5p		hsa-miR-25-5p
					hsa-miR-154-3p		hsa-miR-29a-5p
					hsa-miR-15a-3p		hsa-miR-29b-1-5p
					hsa-miR-16-1-3p		hsa-miR-3176
					hsa-miR-17-5p		hsa-miR-32-3p
					hsa-miR-181b-3p		hsa-miR-362-3p
					hsa-miR-18a-3p		hsa-miR-362-5p
					hsa-miR-18a-5p		hsa-miR-363-3p
					hsa-miR-190b		hsa-miR-3662
					hsa-miR-19a-5p		hsa-miR-429
					hsa-miR-19b-1-5p		hsa-miR-4326
					hsa-miR-200a-3p		hsa-miR-449c-5p
					hsa-miR-200b-3p		hsa-miR-4517
					hsa-miR-20a-3p		hsa-miR-4521
					hsa-miR-215-5p		hsa-miR-484
					hsa-miR-221-5p		hsa-miR-501-5p
					hsa-miR-222-5p		hsa-miR-503-3p
					hsa-miR-25-5p		hsa-miR-503-5p
					hsa-miR-26a-1-3p		hsa-miR-548k
					hsa-miR-29a-5p		hsa-miR-573
					hsa-miR-29b-1-5p		hsa-miR-597-3p
					hsa-miR-3173-5p		hsa-miR-597-5p
					hsa-miR-3176		hsa-miR-6513-3p
					hsa-miR-3187-3p		hsa-miR-7-5p_1
					hsa-miR-32-3p		hsa-miR-7974
					hsa-miR-345-5p		hsa-miR-3934-3p
					hsa-miR-3611		hsa-miR-410-3p
					hsa-miR-362-3p		hsa-let-7b-3p
					hsa-miR-362-5p		hsa-let-7f-2-3p
					hsa-miR-363-3p		hsa-miR-125b-1-3p
					hsa-miR-3662		hsa-miR-1272
					hsa-miR-3684		hsa-miR-128-3p_1
					hsa-miR-424-3p		hsa-miR-16-2-3p
					hsa-miR-429		hsa-miR-17-3p
					hsa-miR-4326		hsa-miR-188-5p
					hsa-miR-449c-5p		hsa-miR-190a-3p
					hsa-miR-4517		hsa-miR-19b-3p
					hsa-miR-4521		hsa-miR-19b-3p_1
					hsa-miR-484		hsa-miR-2278
					hsa-miR-487a-3p		hsa-miR-23a-5p
					hsa-miR-501-5p		hsa-miR-26b-3p
					hsa-miR-502-5p		hsa-miR-30b-3p
					hsa-miR-503-3p		hsa-miR-30d-3p
					hsa-miR-503-5p		hsa-miR-3129-5p
					hsa-miR-548e-3p		hsa-miR-320b_1
					hsa-miR-548k		hsa-miR-331-5p
					hsa-miR-573		hsa-miR-340-3p
					hsa-miR-597-3p		hsa-miR-34c-5p
					hsa-miR-597-5p		hsa-miR-3934-5p
					hsa-miR-6513-3p		hsa-miR-411-3p
					hsa-miR-660-3p		hsa-miR-433-3p
					hsa-miR-744-3p		hsa-miR-449b-5p
					hsa-miR-7-5p_1		hsa-miR-454-5p
					hsa-miR-7974		hsa-miR-487b-5p
					hsa-miR-92a-3p		hsa-miR-495-5p
					hsa-miR-93-3p		hsa-miR-545-5p
							hsa-miR-548n
							hsa-miR-548u
							hsa-miR-561-3p
							hsa-miR-570-3p
							hsa-miR-579-3p
							hsa-miR-580-3p
							hsa-miR-624-3p
							hsa-miR-624-5p
							hsa-miR-627-5p
							hsa-miR-651-3p
							hsa-miR-671-3p
							hsa-miR-6747-3p
							hsa-miR-7-1-3p
							hsa-miR-99a-5p

**Table 4 biomedicines-11-00538-t004:** Intraspecies differentially expressed microRNAs in common.

**Up-Regulated miRNAs in Common**	
MeT10vsMeT5ANT and MeT50vsMeT5ANT	JU10vsJU77NT and JU50vsJU77NT
-	hsa-miR-615-3p
**Down-Regulated miRNAs in Common**	
MeT10vsMeT5ANT and MeT50vsMeT5ANT	JU10vsJU77NT and JU50vsJU77NT
-	hsa-let-7g-3p
	hsa-miR-200b-3p
	hsa-let-7c-3p
	hsa-miR-7974
	hsa-miR-100-3p
	hsa-miR-503-5p
	hsa-miR-222-5p
	hsa-miR-484
	hsa-miR-363-3p
	hsa-miR-4521
	hsa-miR-29a-5p
	hsa-miR-18a-5p
	hsa-miR-29b-1-5p
	hsa-miR-18a-3p
	hsa-miR-503-3p
	hsa-miR-362-5p
	hsa-miR-20a-3p
	hsa-let-7a-2-3p
	hsa-miR-200a-3p
	hsa-miR-132-5p
	hsa-miR-32-3p
	hsa-miR-125b-2-3p
	hsa-miR-16-1-3p
	hsa-miR-19b-1-5p
	hsa-miR-429
	hsa-miR-548k

## Data Availability

Not applicable.

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
