# Peer review of "Small RNA-Seq Transcriptome Profiling of Mesothelial and Mesothelioma Cell Lines Revealed microRNA Dysregulation after Exposure to Asbestos-like Fibers"

_biomedicines, 2023, doi:10.3390/biomedicines11020538_

Round 1
Reviewer 1 Report
The authors performed small RNC-seq transcriptome analysis of miR changes after exposure to Fluoro-edenite, an asbestos-like fibrous ore, in normal pleural mesothelial cells and pleural mesothelioma-derived cell lines in exposed and unexposed situations, respectively.
The results are presented in a comprehensive manner.
The environmental exposure to this ore in Italy is described, which led to the formation of a cluster of pleural malignant mesothelioma.
In the case of environmental exposure, it is assumed that there will be long-term exposure at low concentrations, and one wonders whether the model in this study is suitable for this.
What about the small RNA/miR group extracted in this study, and whether the actual tissues of the clusters have been confirmed in actual cases? 
What is the clinical equivalent of exposure to cells of a malignant mesothelioma cell line?
Normal pleural mesothelial cell lines are very limited in number available, so the current data are not sufficient (even if there is a difference between chronic long-term exposure cases or acute short-term exposure models). However, in the case of cell lines, there are other mesothelioma cell lines available, and I wonder if there is anything that can be learned from the differences in results from cell line to cell line.
Part of the solution to these questions should be added as data.
Author Response
Journal: Biomedicines (ISSN 2227-9059)
Manuscript ID: biomedicines-2164691
Type: Article
Title: Small RNA-Seq transcriptome profiling of mesothelial and mesothelioma cell lines revealed microRNA dysregulation after exposure to asbestos-like fibers
Reviewer 1
The authors performed small RNC-seq transcriptome analysis of miR changes after exposure to Fluoro-edenite, an asbestos-like fibrous ore, in normal pleural mesothelial cells and pleural mesothelioma-derived cell lines in exposed and unexposed situations, respectively.
The results are presented in a comprehensive manner.
The environmental exposure to this ore in Italy is described, which led to the formation of a cluster of pleural malignant mesothelioma.
In the case of environmental exposure, it is assumed that there will be long-term exposure at low concentrations, and one wonders whether the model in this study is suitable for this.
Dear reviewer, what you say is correct. In the case of exposure to asbestos or asbestos-like fibers, the latency period is very long. In fact, the pathology can occur after many years following prolonged exposure to fibers, but also after a single exposure.
In vitro models have the limitation of not being able to reproduce chronic exposure to xenobiotics because after many passages in culture, the cells undergo transformation. However, before carrying out the functional experiments in question, we performed the dose-response curves for both cell lines and we chose the exposure time equal to 48 h because longer exposure times did not lead to differences statistically significant in cell viability. These results have been integrated into the text highlighted in red. New figures 2 and 3 have been added.
What about the small RNA/miR group extracted in this study, and whether the actual tissues of the clusters have been confirmed in actual cases?
In one of our previous study, we validated some microRNAs previously selected in silico by ddPCR in FFPE tissues from some cases of Biancavilla cluster. You can find this work in the bibliography 11 or via this link: https://www.ncbi.nlm.nih.gov/pmc/articles/PMC8618926/ 
Now, our goal is the validation of the most promising results of this study in a subset of patients chronically exposed to fluoro-edenite fibers, using the liquid biopsy, to provide a minimally invasive screening tool for the secondary prevention of malignant pleural mesothelioma. We state this in the Conclusions section.
What is the clinical equivalent of exposure to cells of a malignant mesothelioma cell line?
I believe you wanted to ask what is the clinical equivalent of exposure to fibers of a malignant mesothelioma cell line? The several pathologies correlated with fibers exposure can occur after many years following prolonged exposure to fibers, but also after a single exposure. Epidemiological studies have indeed confirmed that fluoro-edenite fibers have shown similar effects to those already reported after exposure to asbestos fibers (Ledda C, Pomara C, Bracci M, Mangano D, Ricceri V, Musumeci A, Ferrante M, Musumeci G, Loreto C, Fenga C, et al: Natural carcinogenic fiber and pleural plaques assessment in a general population: A cross‑sectional study. Environ Res 150: 23‑29, 2016; Martinez G, Loreto C, Rapisarda V, Musumeci G, Valentino M and Carnazza ML: Effects of exposure to fluoro‑edenite fibre pollution on the respiratory system: An in vivo model. Histol Histopathol 21: 595‑601, 2006) or asbestosis, lung and bronchus cancer, malignant mesothelioma of the pleura, peritoneum, pericardium and tunica vaginalis testis, neoplasms of the ovary, larynx and trachea carcinoma (Aitio A, Cantor KP, Attfield MD, Demers PA, Fowler BA, Grandjean P, Fubini B, Hartwig A, Gérin M, et al: A review of human carcinogens: Arsenic, metals, fibres and dusts. Vol 100 C. IARC, Lyon, pp219‑224, 2012).
Normal pleural mesothelial cell lines are very limited in number available, so the current data are not sufficient (even if there is a difference between chronic long-term exposure cases or acute short-term exposure models). However, in the case of cell lines, there are other mesothelioma cell lines available, and I wonder if there is anything that can be learned from the differences in results from cell line to cell line.
Indeed, there are more commercial mesothelioma cell lines, which in any case are very few compared to other tumor lines. However, it would be interesting to evaluate also potential intercellular differences.
Part of the solution to these questions should be added as data.
As you suggested, the text has been integrated with new data highlighted in red and with the introduction of two new figures.
Reviewer 2
Dear Editor and Authors,
I have reviewed the manuscript titled “Small RNA-Seq transcriptome profiling of mesothelial and mesothelioma cell lines revealed microRNA dysregulation after exposure to asbestos-like fibers” in which the authors investigate the association of biomarkers and specifically miRNA transcriptome in normal and malignant mesothelial lines exposed to and not exposed to asbestos like fibers (FE)!
This is an interesting experimental work. The introduction sets the concept and the focus of the paper quite well and thoroughly. The methodology of the work is precise and appropriate for this type of investigation. The testing used is as expected. The results are interesting, showing that exposure to FE fibers increases the expression of miRNA.
In general this is a good paper, well designed and implemented and well presented. I don’t have any comments and am happy to recommend its publication. Kind regards to all.
Dear reviewer, thank you for the positive comments. We hope to be able to increase our work and to be able to obtain useful results in a clinical setting for this disabling neoplasm.

Reviewer 2 Report
Dear Editor and Authors,
I have reviewed the manuscript titled “Small RNA-Seq transcriptome profiling of mesothelial and mesothelioma cell lines revealed microRNA dysregulation after exposure to asbestos-like fibers” in which the authors investigate the association of biomarkers and specifically miRNA transcriptome in normal and malignant mesothelial lines exposed to and not exposed to asbestos like fibers (FE)!
This is an interesting experimental work. The introduction sets the concept and the focus of the paper quite well and thoroughly. The methodology of the work is precise and appropriate for this type of investigation. The testing used is as expected. The results are interesting, showing that exposure to FE fibers increases the expression of miRNA.
In general this is a good paper, well designed and implemented and well presented. I don’t have any comments and am happy to recommend its publication. Kind regards to all.
Author Response

(The authors gave the same response as above.)

Reviewer 3 Report
In their manuscript entitled “Small RNA-Seq transcriptome profiling of mesothelial and mesothelioma cell lines revealed microRNA dysregulation after exposure to asbestos-like fibers”, Filetti and colleagues evaluated the changes in miRNA transcriptome following exposure of a normal mesothelial cell line and a malignant pleural mesothelioma (MPM) cell line to fluoro-edenite (FE) fibers. The definition of a panel of biomarkers for early diagnosis of MPM patients exposed to asbestos-like fibers would be of great interest. However, the experimental design used by the Authors is inadequate to answer this important question.
What exactly is the message that Authors want to convey? Comparing the effect of FE fibers on a normal mesothelial cell line and a MPM cell line does not make much sense. FE fibers should be the carcinogen that induces the transformation of normal mesothelial cells towards MPM. Thus, to define how exposure to FE fibers impacts on miRNA expression, the only sensible comparison is between untreated normal mesothelial cells and FE fibers-exposed normal mesothelial cells. If the goal, instead, is to answer the question whether miRNAs can be used as biomarkers for early diagnosis, as the Authors wrote in the Conclusions, the expression profile of miRNAs must be studied in a subset of MPM patients exposed to FE fibers by using liquid biopsy. It does not make sense to do it on cell lines, which are useful in mechanistic studies. But if one wants to define a small panel of miRNAs by using cell lines to be validated in a patient cohort, two important points must be considered: (i) comparing untreated normal mesothelial cells to untreated MPM cells would give an idea about miRNAs involved in the transformation process, but the cell line used by the Authors, JU77, was derived by a MPM patient exposed to crocidolite, not FE. Thus, the 333 miRNAs found differentially expressed between MeT5A and JU77 are linked to exposure to crocidolite and give no information regarding FE fibers (maybe the pathogenic process of the latter is the same, maybe not). (ii) Even if the Authors could have used a cell line derived by a MPM patient exposed to FE fibers, this would be not sufficient. How can the Authors exclude that the differences observed between MeT5A and JU77 cells are not cell line-specific? At least 3 normal mesothelial cell lines and 3 MPM cell lines (obviously, exposed to FE fibers and not to canonical asbestos fibers) should be used to drive any conclusion.
From a technical point of view, treatment of cells with FE fibers is unclear. How was the concentration of stock solution determined? Were the fibers simply weighted? Mostly, as FE fibers were obtained from a quarry, how can the Authors guarantee their purity? Were they separated from the raw material present in the quarry? How can the Authors assure that the effects they observed are caused by FE fibers rather than some other contaminating mineral coming from the quarry?
The Authors found more than 100 miRNAs differentially expressed between untreated and FE fibers-exposed JU77 cells. What would be the clinical significance of these miRNAs? They would be modulated only in patients who, after exposure to crocidolite, developed a MPM and after that were exposed also to FE fibers. This would be, to say the least, very bad luck.
The Authors also performed a pathway analysis, which, however, is incomplete. Only (blurred) heatmaps are reported. Where are the statistical significances of pathway deregulations? Moreover, the Authors wrote “MITHrIL fully exploits the topological information encoded by pathways when computing perturbation scores. Pathways are then modeled as complex graphs where each node is a biological element (protein-coding gene, miRNA, or metabolite), and each edge is an interaction between them”. Thus, if each node is a gene, non-coding RNA or metabolite, pathways as reported in the heatmaps should be identified by an additional tool, such as Gene Set Enrichment Analysis. According to the Authors, however, heatmaps are the output of MITHrIL. This part of the manuscript is a long list of pathways that are predicted to be modulated. Many of them (for instance, ovarian steroidogenesis) are clearly irrelevant to mesothelioma biology, but they are indiscriminately reported among others, which are putatively more important (such as AMPK, PI3K-AKT, and Ras). As these are predictions of pathway deregulations inferred simply by differential miRNA expression, at least some of them need to be experimentally validated to have some validity.
Other points:
Lines 107-108: apex is missing.
Table 1: I suppose that column, two by two, represent comparisons, as reported in the text. However, considering that the first row is completely black, it is impossible to understand which column corresponds to which comparison. Moreover, even if the table conveys information regarding the identity of regulated miRNAs, a Venn diagram would be better to visually understand how many regulated miRNAs are in common in different comparison. It is not clear, for instance, whether there is a good overlap between miRNAs regulated by 10 ug/ml FE fibers and miRNAs regulated by 50 ug/ml FE fibers (they cannot be much different).
Lines 201-208 and 215-229: listing here all the regulated pathways does not make much sense.
Author Response
Journal: Biomedicines (ISSN 2227-9059)
Manuscript ID: biomedicines-2164691
Type: Article
Title: Small RNA-Seq transcriptome profiling of mesothelial and mesothelioma cell lines revealed microRNA dysregulation after exposure to asbestos-like fibers
Reviewer 1
The authors performed small RNC-seq transcriptome analysis of miR changes after exposure to Fluoro-edenite, an asbestos-like fibrous ore, in normal pleural mesothelial cells and pleural mesothelioma-derived cell lines in exposed and unexposed situations, respectively.
The results are presented in a comprehensive manner.
The environmental exposure to this ore in Italy is described, which led to the formation of a cluster of pleural malignant mesothelioma.
In the case of environmental exposure, it is assumed that there will be long-term exposure at low concentrations, and one wonders whether the model in this study is suitable for this.
Dear reviewer, what you say is correct. In the case of exposure to asbestos or asbestos-like fibers, the latency period is very long. In fact, the pathology can occur after many years following prolonged exposure to fibers, but also after a single exposure.
In vitro models have the limitation of not being able to reproduce chronic exposure to xenobiotics because after many passages in culture, the cells undergo transformation. However, before carrying out the functional experiments in question, we performed the dose-response curves for both cell lines and we chose the exposure time equal to 48 h because longer exposure times did not lead to differences statistically significant in cell viability. These results have been integrated into the text highlighted in red. New figures 2 and 3 have been added.
What about the small RNA/miR group extracted in this study, and whether the actual tissues of the clusters have been confirmed in actual cases?
In one of our previous study, we validated some microRNAs previously selected in silico by ddPCR in FFPE tissues from some cases of Biancavilla cluster. You can find this work in the bibliography 11 or via this link: https://www.ncbi.nlm.nih.gov/pmc/articles/PMC8618926/ 
Now, our goal is the validation of the most promising results of this study in a subset of patients chronically exposed to fluoro-edenite fibers, using the liquid biopsy, to provide a minimally invasive screening tool for the secondary prevention of malignant pleural mesothelioma. We state this in the Conclusions section.
What is the clinical equivalent of exposure to cells of a malignant mesothelioma cell line?
I believe you wanted to ask what is the clinical equivalent of exposure to fibers of a malignant mesothelioma cell line? The several pathologies correlated with fibers exposure can occur after many years following prolonged exposure to fibers, but also after a single exposure. Epidemiological studies have indeed confirmed that fluoro-edenite fibers have shown similar effects to those already reported after exposure to asbestos fibers (Ledda C, Pomara C, Bracci M, Mangano D, Ricceri V, Musumeci A, Ferrante M, Musumeci G, Loreto C, Fenga C, et al: Natural carcinogenic fiber and pleural plaques assessment in a general population: A cross‑sectional study. Environ Res 150: 23‑29, 2016; Martinez G, Loreto C, Rapisarda V, Musumeci G, Valentino M and Carnazza ML: Effects of exposure to fluoro‑edenite fibre pollution on the respiratory system: An in vivo model. Histol Histopathol 21: 595‑601, 2006) or asbestosis, lung and bronchus cancer, malignant mesothelioma of the pleura, peritoneum, pericardium and tunica vaginalis testis, neoplasms of the ovary, larynx and trachea carcinoma (Aitio A, Cantor KP, Attfield MD, Demers PA, Fowler BA, Grandjean P, Fubini B, Hartwig A, Gérin M, et al: A review of human carcinogens: Arsenic, metals, fibres and dusts. Vol 100 C. IARC, Lyon, pp219‑224, 2012).
Normal pleural mesothelial cell lines are very limited in number available, so the current data are not sufficient (even if there is a difference between chronic long-term exposure cases or acute short-term exposure models). However, in the case of cell lines, there are other mesothelioma cell lines available, and I wonder if there is anything that can be learned from the differences in results from cell line to cell line.
Indeed, there are more commercial mesothelioma cell lines, which in any case are very few compared to other tumor lines. However, it would be interesting to evaluate also potential intercellular differences.
Part of the solution to these questions should be added as data.
As you suggested, the text has been integrated with new data highlighted in red and with the introduction of two new figures.
Reviewer 2
Dear Editor and Authors,
I have reviewed the manuscript titled “Small RNA-Seq transcriptome profiling of mesothelial and mesothelioma cell lines revealed microRNA dysregulation after exposure to asbestos-like fibers” in which the authors investigate the association of biomarkers and specifically miRNA transcriptome in normal and malignant mesothelial lines exposed to and not exposed to asbestos like fibers (FE)!
This is an interesting experimental work. The introduction sets the concept and the focus of the paper quite well and thoroughly. The methodology of the work is precise and appropriate for this type of investigation. The testing used is as expected. The results are interesting, showing that exposure to FE fibers increases the expression of miRNA.
In general this is a good paper, well designed and implemented and well presented. I don’t have any comments and am happy to recommend its publication. Kind regards to all.
Dear reviewer, thank you for the positive comments. We hope to be able to increase our work and to be able to obtain useful results in a clinical setting for this disabling neoplasm.
Reviewer 3
In their manuscript entitled “Small RNA-Seq transcriptome profiling of mesothelial and mesothelioma cell lines revealed microRNA dysregulation after exposure to asbestos-like fibers”, Filetti and colleagues evaluated the changes in miRNA transcriptome following exposure of a normal mesothelial cell line and a malignant pleural mesothelioma (MPM) cell line to fluoro-edenite (FE) fibers. The definition of a panel of biomarkers for early diagnosis of MPM patients exposed to asbestos-like fibers would be of great interest. However, the experimental design used by the Authors is inadequate to answer this important question.
What exactly is the message that Authors want to convey? Comparing the effect of FE fibers on a normal mesothelial cell line and a MPM cell line does not make much sense. FE fibers should be the carcinogen that induces the transformation of normal mesothelial cells towards MPM. Thus, to define how exposure to FE fibers impacts on miRNA expression, the only sensible comparison is between untreated normal mesothelial cells and FE fibers-exposed normal mesothelial cells. If the goal, instead, is to answer the question whether miRNAs can be used as biomarkers for early diagnosis, as the Authors wrote in the Conclusions, the expression profile of miRNAs must be studied in a subset of MPM patients exposed to FE fibers by using liquid biopsy. It does not make sense to do it on cell lines, which are useful in mechanistic studies. But if one wants to define a small panel of miRNAs by using cell lines to be validated in a patient cohort, two important points must be considered: (i) comparing untreated normal mesothelial cells to untreated MPM cells would give an idea about miRNAs involved in the transformation process, but the cell line used by the Authors, JU77, was derived by a MPM patient exposed to crocidolite, not FE. Thus, the 333 miRNAs found differentially expressed between MeT5A and JU77 are linked to exposure to crocidolite and give no information regarding FE fibers (maybe the pathogenic process of the latter is the same, maybe not). (ii) Even if the Authors could have used a cell line derived by a MPM patient exposed to FE fibers, this would be not sufficient. How can the Authors exclude that the differences observed between MeT5A and JU77 cells are not cell line-specific? At least 3 normal mesothelial cell lines and 3 MPM cell lines (obviously, exposed to FE fibers and not to canonical asbestos fibers) should be used to drive any conclusion.
Dear reviewer, thank you for your careful analysis of our manuscript. All the considerations you make are shared by us, but our research group has been dedicating its activity to this topic for many years and unfortunately there are still many limitations to achieve the objectives we have planned. Given that there are very few commercial malignant mesothelioma cell lines, I believe only 2, we have chosen JU77 because they are closest to our model of malignant mesothelioma cells induced by exposure to asbestiform fibers. To date, there are no cell lines from exposure to fluoro-edenite fibers. In this regard, for some time we have been trying to create organoids derived from primary cells of patients with malignant mesothelioma induced by fluoro-edenite fibers, but the cases are rare and the protocols are still being defined. In the meantime, as you suggest and as we stated in the manuscript, we are sampling plasma from malignant mesothelioma cases of Biancavilla and from exposed to fluoro-edenite fibers and unexposed healthy controls. Due to the interspecies difference, it is not new that in vitro experiments are performed between cells derived from the same tumor and non-tumor tissue. To date, unfortunately, there are no commercial malignant mesothelioma cell lines induced by fluoro-edenite fibers, so comparisons between these and multiple healthy mesothelial cell lines cannot be performed.
From a technical point of view, treatment of cells with FE fibers is unclear. How was the concentration of stock solution determined? Were the fibers simply weighted? Mostly, as FE fibers were obtained from a quarry, how can the Authors guarantee their purity? Were they separated from the raw material present in the quarry? How can the Authors assure that the effects they observed are caused by FE fibers rather than some other contaminating mineral coming from the quarry?
The fibers were extracted directly from the ore coming from the quarry in safe conditions. These were sampled using magnifiers, needles and tweezers. Only the fibers were sampled and all traces of soil were removed. Subsequently these were weighed, sterilized by UV light and placed in solution. The concentration of the stock solution was determined by placing a known amount of fibers in a known volume of solution (m/v). This was then sonicated and freshly diluted to obtain the other lower concentration solutions to be used immediately in in vitro experiments. We next tested the morphology of cells in confluent condition immortalized under the microscope and the morphology of single cells immortalized by FlowSight Imaging Flow Cytometer after no exposure and after exposure to fluoro-edenite fibers for both MeT-5A and JU77 cell lines. The results showed the presence of only the fluoro-edenite fibers that pierced the cell membrane and were incorporated into the cell interior. You can find these results in Figure 7 of our previous work at this link https://www.nature.com/articles/s41598-022-13044-0
The Authors found more than 100 miRNAs differentially expressed between untreated and FE fibers-exposed JU77 cells. What would be the clinical significance of these miRNAs? They would be modulated only in patients who, after exposure to crocidolite, developed a MPM and after that were exposed also to FE fibers. This would be, to say the least, very bad luck.
In this phase we preferred not to leave anything out and make all possible comparisons with the data we obtained from the results of the analyses. Subsequently, we will screen and narrow down the panel of microRNAs to be evaluated as you well understood previously.
The Authors also performed a pathway analysis, which, however, is incomplete. Only (blurred) heatmaps are reported. Where are the statistical significances of pathway deregulations? Moreover, the Authors wrote “MITHrIL fully exploits the topological information encoded by pathways when computing perturbation scores. Pathways are then modeled as complex graphs where each node is a biological element (protein-coding gene, miRNA, or metabolite), and each edge is an interaction between them”. Thus, if each node is a gene, non-coding RNA or metabolite, pathways as reported in the heatmaps should be identified by an additional tool, such as Gene Set Enrichment Analysis. According to the Authors, however, heatmaps are the output of MITHrIL. This part of the manuscript is a long list of pathways that are predicted to be modulated. Many of them (for instance, ovarian steroidogenesis) are clearly irrelevant to mesothelioma biology, but they are indiscriminately reported among others, which are putatively more important (such as AMPK, PI3K-AKT, and Ras). As these are predictions of pathway deregulations inferred simply by differential miRNA expression, at least some of them need to be experimentally validated to have some validity.
The heatmap figures have been replaced because they were indeed blurry. As stated in the text, we selected all miRNAs with significant adjusted p-values without a Log2FC cutoff to evaluate the impact of slightly differentially expressed miRNAs on biological pathways. Many of these pathways are apparently irrelevant to the biology of malignant mesothelioma. But, as you well say while some are known to be involved in the pathology, for others it is not. These are however experimental results and will have to be confirmed or excluded by validation.
Other points:
Lines 107-108: apex is missing.
Thank you. We corrected it.
Table 1: I suppose that column, two by two, represent comparisons, as reported in the text. However, considering that the first row is completely black, it is impossible to understand which column corresponds to which comparison. Moreover, even if the table conveys information regarding the identity of regulated miRNAs, a Venn diagram would be better to visually understand how many regulated miRNAs are in common in different comparison. It is not clear, for instance, whether there is a good overlap between miRNAs regulated by 10 ug/ml FE fibers and miRNAs regulated by 50 ug/ml FE fibers (they cannot be much different).
You have guessed well from the text. The chosen table style hid the header line. I corrected the table by changing the style and now the header is readable. Thanks so much for the alert.
Lines 201-208 and 215-229: listing here all the regulated pathways does not make much sense.
We have preferred to insert these lists in the text to make this information available to readers without having to read the heatmaps which are not easily accessible to everyone.
Round 2
Reviewer 1 Report
Although this model is lkind of far from clinical and environmental exposure, the experiments were seriously performed. Authors responded to the reviewer's comments and added new data. Thus, this revised manuscript seemed to be OK.
Author Response
Dear Reviewer, thank you for your appreciation and support in improving the manuscript.
Reviewer 3 Report
Given many issues researchers must deal with when working on mesothelioma, I understand that Authors could not perform several experiments that would be important to achieve their objectives. About 10 different MPM cell lines are commercially available, not only 2, but anyway none of them is representative of MPM patients exposed to fluoro-edenite fibers. In this regard, the attempt of the Authors to obtain organoids from patients with mesothelioma caused by fluoro-edenite fibers is highly interesting, although challenging.
I checked the previous work from the Authors (https://www.nature.com/articles/s41598-022-13044-0), which, in the present manuscript, is only cited in the Materials and Methods section, and now I am wondering: what would this new manuscript add to that work? They both used the same techniques on the same cell lines, to study the miRNA transcriptome following exposure to the same doses of fluoro-edenite fibers. Let’s take Figures 5 and 6 of the previous work and compare them with Figures 4 and 5 of the present one: the same comparisons are considered to do the same pathway analysis, with results largely overlapping.
Author Response
Given many issues researchers must deal with when working on mesothelioma, I understand that Authors could not perform several experiments that would be important to achieve their objectives. About 10 different MPM cell lines are commercially available, not only 2, but anyway none of them is representative of MPM patients exposed to fluoro-edenite fibers. In this regard, the attempt of the Authors to obtain organoids from patients with mesothelioma caused by fluoro-edenite fibers is highly interesting, although challenging.
Dear Reviewer, I'm glad you understand the difficulty of this area of research. However we will always try to improve and reach important goals in the clinical field. In any case, I told you that there are very few commercial cell lines that respond to our case because they must have the characteristics of being human, adherent, epithelial-like and specific to mesothelioma. To date, those that meet these requirements are: JU77, LO68, H2052, and Mero.
I checked the previous work from the Authors (https://www.nature.com/articles/s41598-022-13044-0), which, in the present manuscript, is only cited in the Materials and Methods section, and now I am wondering: what would this new manuscript add to that work? They both used the same techniques on the same cell lines, to study the miRNA transcriptome following exposure to the same doses of fluoro-edenite fibers. Let’s take Figures 5 and 6 of the previous work and compare them with Figures 4 and 5 of the present one: the same comparisons are considered to do the same pathway analysis, with results largely overlapping.
Dear reviewer, the data that come out of the results of a sequencing are really a lot. In the previous work that I myself reported to you, we carried out a different data analysis and you can see this from the workflow of both works (Figure 1). Furthermore, in the previous manuscript we addressed both microRNAs and dysregulated tRNA-derived ncRNAs. In this work we are already at the next step, in fact we have focused on microRNAs to select the panel that will be validated by ddPCR on the plasma samples we are collecting. These results will be in the next work.
Round 3
Reviewer 3 Report
None.